# Determining Layer-wise Sparsity for Large Language Models Through a Theoretical Perspective

**Weizhong Huang** [1]  **Yuxin Zhang** [1]  **Xiawu Zheng** [1 2 3]  **Fei Chao** [1]  **Rongrong Ji** [1 2]

## Abstract

In this paper, we address the challenge of determining the layer-wise sparsity rates of large language models (LLMs) through a theoretical perspective. Specifically, we identify a critical issue of **"reconstruction error explosion"** in existing LLMs sparsification methods. This refers to the cumulative effect of reconstruction errors throughout the sparsification process, where errors from earlier layers propagate and amplify in subsequent layers. As a result, the overall reconstruction error increases significantly, leading to a substantial degradation in model performance. Through theoretical analysis, we derive a simple yet effective approach to layer-wise sparsity allocation that mitigates this issue. Our method uses a monotonically increasing arithmetic progression, reducing the process of determining sparsity rates for multiple layers to the determination of a single common difference hyperparameter. Remarkably, this allows for the optimal layer-wise sparsity rates to be identified with just a few trials. Both our theoretical analysis and experimental results demonstrate that this sparsity allocation scheme is near optimal. Extensive experiments show that our method significantly improves the performance of sparse LLMs across various architectures, outperforming existing layer-wise sparsity methods. Furthermore, it enhances the performance of various compression techniques and is applicable to vision and multimodal models. Notably, our method achieves a reduction of 52.10 in perplexity for the 70% sparse LLaMA2-7B model obtained via Wanda, improves average zero-shot accuracy by 10.50%, and deliv-

ers speedups of $2.63\times$ and $2.23\times$ on CPU and GPU, respectively. Code is available at https://github.com/wzhuang-xmu/ATP.

## 1. Introduction

Large Language Models (LLMs) have demonstrated outstanding capabilities in various natural language processing tasks (Meta, 2024; Yang et al., 2024; Liu et al., 2024a). However, their vast number of parameters and high computational demands present significant challenges to model deployment, impeding further applications (Zhu et al., 2024; Wang et al., 2024). Network sparsity (Rao et al., 2021; Paul et al., 2022; Huang et al., 2025b) methods remove less important parameters from LLMs, enabling model compression without sacrificing performance. This can reduce the model's memory footprint and computational complexity (Li et al., 2024a; An et al., 2024). Existing sparsity methods for LLMs, such as SparseGPT (Frantar & Alistarh, 2023) and Wanda (Sun et al., 2023), adopt a post-training approach which prune all weights in one-shot and can obtain sparse LLMs without the need for additional fine-tuning.

However, these sparsity methods set a uniform layer-wise sparsity rate for different layers, without considering the varying importance of each layer, which harms the accuracy of sparse LLMs. To address the above issue, many studies have proposed various methods to determine the layer-wise sparsity rate of LLMs. Based on their methodological designs, we categorize these methods into two main groups:

**Metric based methods.** These methods determine the importance of each layer of LLMs through hand-crafted metrics, thereby obtaining the sparsity rate of each layer. For example, OWL (Yin et al., 2023) proposes an outlier weighted layer importance metric. By setting the sparsity rate to be inversely proportional to the outlier ratio, it effectively protects the layers with a higher ratio of outliers. AlphaPruning (Lu et al., 2024) utilizes the heavy-tailed self- regularization theory (Martin & Mahoney, 2019), especially the shape of the empirical spectral density (Martin et al., 2021) of the weight matrix, to determine the importance of each layer of LLMs. ALS (Li et al., 2024c) proposes an importance metric based on mutual information (Tschannen et al., 2019),

[1] Key Laboratory of Multimedia Trusted Perception and Efficient Computing, Ministry of Education of China, Xiamen University, 361005, P.R. China. [2] Institute of Artificial Intelligence, Xiamen University. [3] Peng Cheng Laboratory, Shenzhen, China. Correspondence to: Rongrong Ji <rrji@xmu.edu.cn>.

*Proceedings of the 42nd International Conference on Machine Learning*, Vancouver, Canada. PMLR 267, 2025. Copyright 2025 by the author(s).

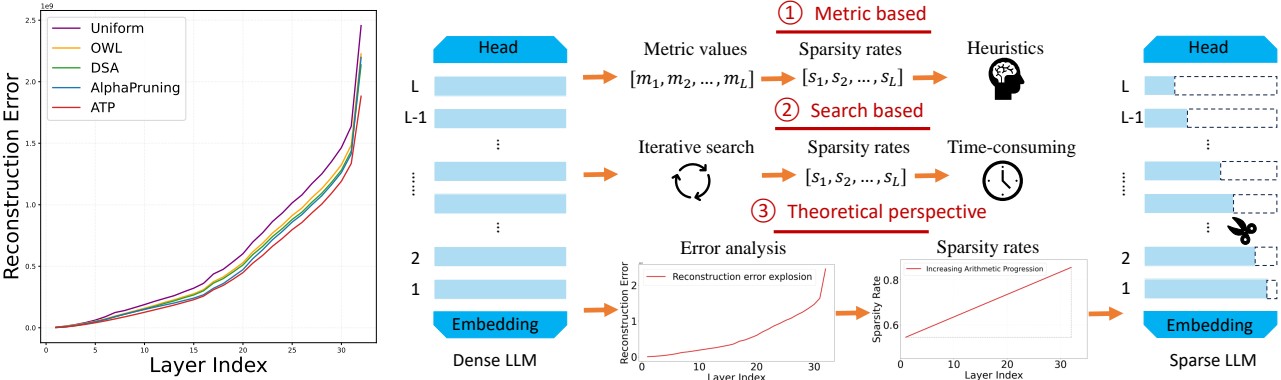

*Figure 1.* (Left) shows the comparison of reconstruction error among different layer-wise sparsity methods. All methods face the problem of **"reconstruction error explosion"**; however, our method achieves lower reconstruction error compared to other methods. (Right) presents a comparison between our method and other layer-wise sparsity methods. The metric-based method calculates the importance of each layer to obtain the sparsity rate. However, this method is heuristically designed by human experts and is not optimal. And the search-based method requires a large number of iterative searches, which is time-consuming. In contrast, we analyze the causes of **"reconstruction error explosion"** from a theoretical perspective, and deduce theoretically that using a monotonically increasing arithmetic progression to determine the layer-wise sparsity rate can alleviate the problem of **"reconstruction error explosion"**.

and sets a higher sparsity rate for layers with higher mutual information. Although these metrics have proven their effectiveness experimentally, manually designing metric requires extensive validation and complex calculations are needed to obtain the sparsity rate of each layer. Most importantly, most of these methods lack theoretical analysis, making it impossible to ensure that the solutions obtained are optimal.

**Search based methods.** In addition to these heuristics designed by humans, recently, there have also been some methods that adopt a search-based approach to determine the layer-wise sparsity rate of LLMs. For example, DSA (Li et al., 2024b) develops an expression discovery framework to explore potential sparsity rate allocation strategies and obtains layer-wise sparsity allocation function by searching. However, the evolutionary search method employed by DSA requires 2000 iterations. For large-scale LLMs with a vast number of parameters, this demands a search process lasting several days, which incurs a significant cost. In addition, in order to obtain the final allocation function, DSA designs a complex search space and process, which means the effectiveness of the method heavily depends on the experience of human experts.

In this paper, we rethink the approach of determining the layer-wise sparsity rate of LLMs, and derive the layer-wise sparsity rate of LLMs from reconstruction error perspective. Specifically, we first prove that increasing the sparsity rate leads to an increase in the reconstruction error of the corresponding layer. Additionally, we show that an increase in the reconstruction error of one layer causes an increase in the reconstruction error of subsequent layers. This implies that increasing the sparsity rate of earlier layers not only increases the reconstruction error of the corresponding

layer, but also leads to an increase in the reconstruction errors of all subsequent layers. As the network propagates forward, the reconstruction errors accumulate, causing the total reconstruction error to grow significantly, thus causing **"reconstruction error explosion"** (See the left in Figure 1).

Through the above theoretical analysis, we provide a simple yet effective rule for determining the layer-wise sparsity rates of LLMs: *the sparsity rate should be lower in earlier layers, and the layer-wise sparsity rates should follow a increasing pattern*. This approach effectively alleviates the issue of **"reconstruction error explosion"**, resulting in a well-performing sparse LLM. To achieve this, we use a monotonically increasing arithmetic progression to determine the sparsity rates for all layers of LLMs and employ grid search to find the common difference of the arithmetic progression. Since the range of valid values for the common difference is narrow, our search is highly efficient, and after only a few attempts, we can determine the common difference that yields the best accuracy.

Furthermore, we prove that the total reconstruction error obtained from the monotonically increasing sparsity scheme is less than that of any non-monotonically increasing sparsity scheme. This indicates that our method is theoretically close to the optimal solution. Additionally, we compare our sparsity rate scheme with the optimal solution obtained through Bayesian search, we find that our scheme is close to the optimal solution found by search. This indicates that our method is empirically close to the optimal solution.

To evaluate the effectiveness of our ATP [1] method, we con-

---

[1] Determining layer-wise sparsity for LLMs through **A** **T**heoretical **P**erspective (**ATP**).

duct extensive experiments on LLMs of various architectures, with parameter counts ranging from 6.7 billion to 70 billion. Our evaluation metrics include perplexity, average accuracy across seven zero-shot datasets, and performance on arithmetic and knowledge reasoning tasks. Our ATP method demonstrate substantial improvements over existing post-training sparsity techniques, significantly surpassing other layer-wise sparsity methods. Notably, ATP reduce the perplexity of the 70% sparse LLaMA2-7B pruned using Wanda (Sun et al., 2023) by 52.10 and increase average zero-shot accuracy by 10.50%, outperforming the state-of-the-art AlphaPruning method (Lu et al., 2024) by 6.71 and 2.46%, respectively. Additionally, ATP achieved 2.63× and 2.23× speedups on CPU and GPU, respectively, and require only 18 minutes to compute layer-wise sparsity rates. Furthermore, we evaluate ATP's enhancements on various compression techniques, including N:M sparsity, structured pruning, and network quantization, as well as its benefits for sparse multimodal and sparse vision models. These experimental results clearly demonstrate that ATP provides substantial performance improvements for compressed models.

## 2. Related Work

**LLMs Sparsity.** Before the advent of LLM, a variety of sparsity techniques had been developed to compress models such as ResNet (Yu et al., 2022b; Zhang et al., 2024) , BERT (Xia et al., 2022; Li et al., 2023), and ViT (Yu et al., 2022a; He et al., 2024). Meanwhile, researchers have developed several post-training sparsity methods specifically for LLMs. For example, SparseGPT (Frantar & Alistarh, 2023) uses the inverse of the Hessian matrix for pruning and pruned weight updates. Wanda (Sun et al., 2023) uses a metric that combines weight magnitude and input activation to prune LLMs, while Pruner-zero (Dong et al., 2024) searches for symbolic pruning metric using genetic programming. Additionally, ALPS (Meng et al., 2024) uses an Alternating Direction Method of Multiplier (ADMM) (Boyd et al., 2011)-based approach to prune LLMs in one-shot. The above methods focus on determining the mask within the layer of LLMs and setting a uniform layer-wise sparsity rate. Our work study the layer-wise sparsity allocation problem in sparse LLMs from the perspective of reconstruction error, thereby effectively improving the accuracy of above methods.

**Layer-wise Sparsity.** Layer-wise sparsity rate determines the number of weights to be retained in each layer of the network (Lee et al., 2020; Frankle et al., 2020; Liu et al., 2022a; Huang et al., 2025a). To determine the layer-wise sparsity rate in LLMs, OWL(Yin et al., 2023) proposes an outlier-weighted metric, Alphapruning (Lu et al., 2024) uses heavy-tailed self-regularization theory (Martin & Mahoney, 2019), ALS (Li et al., 2024c) proposes a layer redundancy metric based on mutual information (Kraskov et al., 2004),

DSA (Li et al., 2024b) develops an expression discovery algorithm to explore potential sparsity allocation. However, the above metric and search based methods all lack theoretical proof of effectiveness and require complex calculations or search to obtain layer-wise sparsity rate. In comparison, our method directly derives the layer-wise sparsity rates based on a monotonically increasing arithmetic progression from the perspective of reconstruction error, requiring only the determination of the common difference to quickly obtain the sparsity rate for each layer. More importantly, we have validated the effectiveness of the above method through rigorous proof.

**Reconstruction Error.** Reconstruction error is a metric for measuring the difference between the output of a compressed network and that of the original network. A smaller reconstruction error generally implies that the compressed network can better preserve the performance of the original network (Yun & Wong, 2021; Hubara et al., 2021; Ma et al., 2023b). In order to minimize the reconstruction error of sparse LLMs, SparseGPT (Frantar & Alistarh, 2023) proposes a mask selection and weight update algorithm based on Hessian inverse and DSnoT (Zhang et al., 2023) proposes a training-free dynamic weight pruning and growing algorithm. In this paper, we explore the **"reconstruction error explosion"** problem in sparse LLMs. Specifically, the reconstruction error accumulates and magnifies across layers, leading to an extremely large overall reconstruction error, which undermines the model's accuracy. Therefore, we propose our layer-wise sparsity allocation method to alleviate the above **"reconstruction error explosion"** problem.

## 3. Methodology

**Notation.** In this paper, we use bold typeface indicates matrices (*e.g.*, $\boldsymbol{W}, \boldsymbol{X}$) and calligraphic font represents loss functions or models (*e.g.*, $\mathcal{L}, \mathcal{M}$).

### 3.1. Preliminaries

Without loss of generality, we take each layer in the LLMs as the basic unit of analysis. These layers contain modules such as Attention (Vaswani et al., 2017), MLP (Popescu et al., 2009), LayerNorm (Lei Ba et al., 2016), and residual connections (He et al., 2016), etc. We represent a layer's computation as $\boldsymbol{W}\boldsymbol{X}$, where $\boldsymbol{W}$ is the layer's weight and $\boldsymbol{X}$ is the input. Consider an LLM composed of $L$ layers, we define the reconstruction error of the $i$-th layer ($i = 1, 2, \cdots, L$) as follows:

$$\mathcal{L}(\boldsymbol{W}_i, \boldsymbol{X}_i) = \left\| \boldsymbol{W}_i \boldsymbol{X}_i - \widetilde{\boldsymbol{W}}_i \widetilde{\boldsymbol{X}}_i \right\|_F^2 \qquad (1)$$

where $\boldsymbol{W}_i \in \mathbb{R}^{c_{out} \times c_{in}}$ and $\boldsymbol{X}_i \in \mathbb{R}^{c_{in} \times d}$ are the weights and input of the $i$-th layer respectively. $\widetilde{\boldsymbol{W}}_i$ and $\widetilde{\boldsymbol{X}}_i$ are the

corresponding sparse versions, where $c_{in}$ and $c_{out}$ represent the number of input and output feature dimensions, $d$ is the hidden dimension, and $\|\cdot\|_F$ is Frobenius norm.

Existing post-training sparsity methods all attempt to minimize the reconstruction error of sparse LLMs (*e.g.*, SparseGPT and Wanda) and a large amount of experimental evidence in the these papers indicates that a sparse LLM with good accuracy often has a low reconstruction error.

In the next section, we reveal that the existing post-training sparsity methods all have the problem of **"reconstruction error explosion"**.

### 3.2. "Reconstruction Error Explosion" in Sparse LLMs

First, we analyze the relationship between the sparsity rate and the reconstruction error. Briefly, a higher sparsity rate leads to a higher reconstruction error. Formally, we propose Theorem 3.1.

**Theorem 3.1** (**Effect of increased sparsity on reconstruction error**). *When the input is the same, increasing the sparsity of the weights in the $i$-th layer will lead to an increase in the reconstruction error of this layer.*

*Proof of Theorem 3.1.* We only consider the effect of sparse weights on the reconstruction error, and we ignore the error caused by the input at this time. Therefore, we consider the impact of the sparse weights of the $i$-th layer on the reconstruction error when the input is the same. That is, $\widetilde{\boldsymbol{X}}_i = \boldsymbol{X}_i$. Then, the reconstruction error of the $i$-th layer is expressed as:

$$\mathcal{L} = \left\| \boldsymbol{W}_i \boldsymbol{X}_i - \widetilde{\boldsymbol{W}}_i \boldsymbol{X}_i \right\|_F^2 \tag{2}$$

Consider two weights $\widetilde{\boldsymbol{W}}_i^{(1)}$ and $\widetilde{\boldsymbol{W}}_i^{(2)}$ of different sparsity, such that $\widetilde{\boldsymbol{W}}_i^{(1)}$ has lower sparsity (i.e., has fewer zero elements). The difference in reconstruction error corresponding to these two sparse weights is:

$$
\begin{aligned}
\mathcal{L}^{(1)} - \mathcal{L}^{(2)} &= \left\| \boldsymbol{W}_i \boldsymbol{X}_i - \widetilde{\boldsymbol{W}}_i^{(1)} \boldsymbol{X}_i \right\|_F^2 \\
&- \left\| \boldsymbol{W}_i \boldsymbol{X}_i - \widetilde{\boldsymbol{W}}_i^{(2)} \boldsymbol{X}_i \right\|_F^2 = \left\| \boldsymbol{W}_i \boldsymbol{X}_i - \widetilde{\boldsymbol{W}}_i^{(1)} \boldsymbol{X}_i \right\|_F^2 \\
&- \left\| (\boldsymbol{W}_i \boldsymbol{X}_i - \widetilde{\boldsymbol{W}}_i^{(1)} \boldsymbol{X}_i) + (\widetilde{\boldsymbol{W}}_i^{(1)} \boldsymbol{X}_i - \widetilde{\boldsymbol{W}}_i^{(2)} \boldsymbol{X}_i) \right\|_F^2 \\
&= -2\langle (\boldsymbol{W}_i - \widetilde{\boldsymbol{W}}_i^{(1)}) \boldsymbol{X}_i, (\widetilde{\boldsymbol{W}}_i^{(1)} - \widetilde{\boldsymbol{W}}_i^{(2)}) \boldsymbol{X}_i \rangle_F \\
&- \left\| \widetilde{\boldsymbol{W}}_i^{(1)} \boldsymbol{X}_i - \widetilde{\boldsymbol{W}}_i^{(2)} \boldsymbol{X}_i \right\|_F^2
\end{aligned}
\tag{3}
$$

The first term of the inner product, $(\boldsymbol{W}_i - \widetilde{\boldsymbol{W}}_i^{(1)}) \boldsymbol{X}_i$, represents the error introduced by sparsifying the orig-

inal weight matrix $\boldsymbol{W}_i$ to obtain a less sparse version $\widetilde{\boldsymbol{W}}_i^{(1)}$. The second term, $(\widetilde{\boldsymbol{W}}_i^{(1)} - \widetilde{\boldsymbol{W}}_i^{(2)}) \boldsymbol{X}_i$, quantifies the additional error resulting from further sparsifying $\widetilde{\boldsymbol{W}}_i^{(1)}$ to derive an even sparser matrix $\widetilde{\boldsymbol{W}}_i^{(2)}$. Since both errors $(\boldsymbol{W}_i - \widetilde{\boldsymbol{W}}_i^{(1)}) \boldsymbol{X}_i$ and $(\widetilde{\boldsymbol{W}}_i^{(1)} - \widetilde{\boldsymbol{W}}_i^{(2)}) \boldsymbol{X}_i$ are generated through the same sparsification method, they point in similar directions within the vector space. This alignment ensures that their Frobenius inner product satisfies $\langle (\boldsymbol{W}_i - \widetilde{\boldsymbol{W}}_i^{(1)}) \boldsymbol{X}_i, (\widetilde{\boldsymbol{W}}_i^{(1)} - \widetilde{\boldsymbol{W}}_i^{(2)}) \boldsymbol{X}_i \rangle_F > 0$. And since $\left\| \widetilde{\boldsymbol{W}}_i^{(1)} \boldsymbol{X}_i - \widetilde{\boldsymbol{W}}_i^{(2)} \boldsymbol{X}_i \right\|_F^2 > 0$, therefore:

$$\mathcal{L}^{(1)} - \mathcal{L}^{(2)} < 0 \tag{4}$$

Therefore, the reconstruction error $\mathcal{L}^{(1)}$ for lower sparsity is smaller than $\mathcal{L}^{(2)}$ for higher sparsity. In other words, increasing the sparsity will lead to an increase in the reconstruction error of this layer. So, we have completed the proof of the above theorem. $\square$

On the other hand, we find that there is an cumulative effect of reconstruction error during the sparsification process. It is manifested as the reconstruction error in LLMs showing an increasing trend due to the influence of previous sparse layers. In other words, if the reconstruction error of the previous layer increases, the reconstruction error of the subsequent layers will also increase accordingly. Formally, we propose Theorem 3.2.

**Theorem 3.2** (**The cumulative effect of reconstruction error**). *When the reconstruction error of the $i$-th layer increases, it leads to an increase in the lower bound of the reconstruction error for the $(i + 1)$-th layer.*

To prove Theorem 3.2, we need to define the following lemma:

**Lemma 3.3.** *Let $\boldsymbol{A} \in \mathbb{R}^{m \times n}$ and $\boldsymbol{B} \in \mathbb{R}^{n \times p}$ be arbitrary matrices. Then, it holds that $\|\boldsymbol{AB}\|_F^2 \geq \sigma_{\min}^2(\boldsymbol{A}) \|\boldsymbol{B}\|_F^2$, where $\sigma_{\min}(\boldsymbol{A})$ denotes the smallest non-zero singular value of $\boldsymbol{A}$.*

The proof of the Lemma 3.3 can be found in Appendix B. Now we formally prove Theorem 3.2.

*Proof of Theorem 3.2.* The reconstruction error for the $(i + 1)$-th layer can be expressed as:

$$
\begin{aligned}
\mathcal{L}(\boldsymbol{W}_{i+1}, \boldsymbol{X}_{i+1}) &= \left\| \boldsymbol{W}_{i+1} \boldsymbol{X}_{i+1} - \widetilde{\boldsymbol{W}}_{i+1} \widetilde{\boldsymbol{X}}_{i+1} \right\|_F^2 \\
&= \left\| (\boldsymbol{W}_{i+1} - \widetilde{\boldsymbol{W}}_{i+1}) \boldsymbol{X}_{i+1} + \widetilde{\boldsymbol{W}}_{i+1} (\boldsymbol{X}_{i+1} - \widetilde{\boldsymbol{X}}_{i+1}) \right\|_F^2
\end{aligned}
$$

$$= \left\| (\boldsymbol{W}_{i+1} - \widetilde{\boldsymbol{W}}_{i+1})\boldsymbol{X}_{i+1} \right\|_F^2$$
$$+ \left\| \widetilde{\boldsymbol{W}}_{i+1}(\boldsymbol{X}_{i+1} - \widetilde{\boldsymbol{X}}_{i+1}) \right\|_F^2$$
$$+ 2\mathrm{tr}((\boldsymbol{W}_{i+1} - \widetilde{\boldsymbol{W}}_{i+1})\boldsymbol{X}_{i+1})^\top (\widetilde{\boldsymbol{W}}_{i+1}(\boldsymbol{X}_{i+1} - \widetilde{\boldsymbol{X}}_{i+1})) \tag{5}$$

Since the third term is the trace of the product of two matrices, the first and second term is the square of the Frobenius norm, and considering that the weights and inputs in LLMs are matrices with large dimensions, the magnitude of the first and second term is much greater than the third term, so we ignore the third term and get:

$$\mathcal{L}(\boldsymbol{W}_{i+1}, \boldsymbol{X}_{i+1}) \approx \left\| (\boldsymbol{W}_{i+1} - \widetilde{\boldsymbol{W}}_{i+1})\boldsymbol{X}_{i+1} \right\|_F^2$$
$$+ \left\| \widetilde{\boldsymbol{W}}_{i+1}(\boldsymbol{X}_{i+1} - \widetilde{\boldsymbol{X}}_{i+1}) \right\|_F^2 \tag{6}$$

Since $\left\| (\boldsymbol{W}_{i+1} - \widetilde{\boldsymbol{W}}_{i+1})\boldsymbol{X}_{i+1} \right\|_F^2 > 0$. Therefore:

$$\mathcal{L}(\boldsymbol{W}_{i+1}, \boldsymbol{X}_{i+1}) > \left\| \widetilde{\boldsymbol{W}}_{i+1}(\boldsymbol{X}_{i+1} - \widetilde{\boldsymbol{X}}_{i+1}) \right\|_F^2 \tag{7}$$

According to Lemma 3.3, we get:

$$\mathcal{L}(\boldsymbol{W}_{i+1}, \boldsymbol{X}_{i+1}) > \sigma_{\min}^2(\widetilde{\boldsymbol{W}}_{i+1}) \left\| (\boldsymbol{X}_{i+1} - \widetilde{\boldsymbol{X}}_{i+1}) \right\|_F^2$$
$$= \sigma_{\min}^2(\widetilde{\boldsymbol{W}}_{i+1}) \left\| (\boldsymbol{W}_i \boldsymbol{X}_i - \widetilde{\boldsymbol{W}}_i \widetilde{\boldsymbol{X}}_i) \right\|_F^2$$
$$= \sigma_{\min}^2(\widetilde{\boldsymbol{W}}_{i+1}) \mathcal{L}(\boldsymbol{W}_i, \boldsymbol{X}_i) \tag{8}$$

Since $\sigma_{\min}^2(\widetilde{\boldsymbol{W}}_{i+1}) > 0$, we have proven that the increase of the reconstruction error of the $i$-th layer will lead to the increase of the lower bound of the reconstruction error of the $(i + 1)$-th layer. $\quad\square$

Theorem 3.2 shows that an increase in the reconstruction error of the previous layer in a sparse LLM usually leads to a further increase in the lower bound of the reconstruction error of the subsequent layer. In practice, this often means that an increase in the reconstruction error of the previous layer will lead to an increase in the reconstruction error of the subsequent layer. We have also observed this phenomenon in the left of Figure 1. We can see that when the reconstruction error of the earlier layers is smaller, the reconstruction error of the subsequent layers is also smaller. Conversely, when the reconstruction error of the earlier layers is larger, the reconstruction error of the subsequent layers is also larger.

According to Theorems 3.1 and 3.2, we can easily get the following Theorem 3.4:

**Theorem 3.4** (**Impact of the sparsity of the previous layer on the reconstruction error of the next layer.**). *Increasing the sparsity of the $i$-th layer will lead to an increase in the lower bound of the reconstruction error of the $(i + 1)$-th layer.*

*Proof of Theorem 3.4.* According to Theorems 3.1 and 3.2, we have the following relationship:

$$\text{sparsity of layer } i \uparrow \Longrightarrow \mathcal{L}(\boldsymbol{W}_i, \boldsymbol{X}_i) \uparrow$$
$$\Longrightarrow \mathcal{L}(\boldsymbol{W}_{i+1}, \boldsymbol{X}_{i+1}) > \sigma_{\min}^2(\widetilde{\boldsymbol{W}}_{i+1})\mathcal{L}(\boldsymbol{W}_i, \boldsymbol{X}_i) \uparrow . \tag{9}$$

Therefore, we complete the proof of Theorem 3.4. $\quad\square$

To summarize all the above, we can get the following logical chain of our paper: sparsity rate of 1-st layer $\propto$ reconstruction error of 1-st layer $\propto$ reconstruction error of $L$-th layer $\propto$ total error $\propto$ accuracy loss. That is, when the sparsity rate of the 1-st layer increases, it will lead to an increase in the reconstruction error of this layer. As the reconstruction error accumulates continuously from the 1-st layer to the $L$-th layer, the reconstruction error of the $L$-th layer also increases accordingly. Then, the reconstruction errors of all layers of the model will exhibit an explosion phenomenon, resulting in a serious decline in the accuracy of the sparse model. We refer to the above phenomenon as the **"reconstruction error explosion"**. This phenomenon can be observed on the left side of Figure 1.

From the above theoretical analysis, we understand that the earlier layers are more important than the later layers. Setting a lower sparsity rate for the previous layers helps alleviate the problem of **"reconstruction error explosion"**. In the next section, we will introduce our method of determining the layer-wise sparsity rate in detail.

### 3.3. Discussion about the Rationality of Theoretical Modeling

In Sec. 3.1, we represent a Transformer layer's computation as $\boldsymbol{W}\boldsymbol{X}$. The Transformer layer includes components such as Attention, MLP, nonlinearities, and layer normalization. Due to the more complex nonlinear calculations in a Transformer layer, there are differences between theoretical analysis based on $\boldsymbol{W}\boldsymbol{X}$ and the actual architecture. However, our analysis remains reasonable for the following reasons:

1. **The theoretical modeling of $\boldsymbol{W}\boldsymbol{X}$ is sufficient to analyze the layer's reconstruction error.** Our method sparsifies the linear layers in Attention and MLP modules, while other components remain unaffected. These linear layers account for the majority of the parameter count and significantly influence the computation results of the layer. The sparsified linear layers dominate the computation of the reconstruction error for each layer. Although various nonlinear operations exist in the actual architecture, they typically do not fundamentally alter the reconstruction error of each layer and have minimal impact on theoretical analysis. Therefore, modeling the primary computation of a layer as

$\boldsymbol{WX}$ is sufficient for analyzing the reconstruction error of that layer. This is also sufficient for us to analyze how reconstruction errors accumulate and propagate across the network.

2. **Transformer's linear modeling is supported by existing research.** Razzhigaev et al. (Razzhigaev et al., 2024) employs Procrustes similarity analysis to discover that the embedding transformations between sequential layers in LLMs such as GPT, LLaMA, OPT, and BLOOM exhibit a near-perfect linear relationship, with linear score is 0.99. This indicates that despite the non-linear operations within Transformer layers, the mapping between adjacent layers can still be approximated as a linear transformation. Therefore, it reasonable and natural to model Transformer layer computations using $\boldsymbol{WX}$.

3. **Modeling the computation of modules as $\boldsymbol{WX}$ is a common practice in many works.** AdaRound (Nagel et al., 2020) and MRECG (Ma et al., 2023b) simplify the computation of the CONV+BatchNorm+RELU modules in quantized convolutional neural networks as $\boldsymbol{WX}$ when analyzing reconstruction error. This approach of ignoring unnecessary computations and focusing on the core computations is a common practice, which facilitates the derivation of theoretical results.

### 3.4. Determining Layer-wise Sparsity Rates for LLMs

According to the theorem in Sec. 3.2, the reconstruction error in sparse LLMs have **"reconstruction error explosion"** problem. Specifically, the error from the earlier layers will cause an increase in the error of the later layers. When the error from all earlier layers are accumulated, it will lead to a sharp increase in the error of the later layers. Meanwhile, this can lead to an increase in the total reconstruction error of sparse LLMs, thereby damaging the final accuracy of the sparse LLMs. Therefore, in order to mitigate the negative impact of the **"reconstruction error explosion"** of reconstruction errors on sparse LLMs, we can set the earlier layers to have lower sparsity and the later layers to have higher sparsity. Therefore, we propose to determine the sparsity rate of each layer in LLMs according to the following monotonically increasing arithmetic progression:

$$s_i = S - \frac{\beta(L-1)}{2} + \beta \times (i-1), \quad i = 1, 2, \ldots, L \quad (10)$$

where $s_i$ is the sparsity rate (fraction of zero entries) of the $i$-th layer, $L$ is the total layer number of LLMs, and $S$ is the average sparsity rate of all layers. $\beta$ is a hyperparameter that controls the degree of difference in the sparsity rate of each layer of LLMs. The above formula means that we only need to determine the hyperparameter $\beta$ to get the sparsity rate of each layer of LLMs.

We use grid search (Jiménez et al., 2008) to determine $\beta$ for sparse LLMs. Specifically, since $0 \leq s_0, s_L \leq 1$ and considering the relatively low sparsity rate set for the earlier layers, the arithmetic progression should be increasing. Therefore, we can deduce that the possible range of values for $\beta$ is $0 < \beta \leq \min(\frac{2S}{L-1}, \frac{2(1-S)}{L-1})$. This is a small range for a sparse LLMs. For example, for a LLaMA3-8B (Meta, 2024) model with 32 layers and the average sparsity rate is $S = 0.7$, then the range is $0 < \beta \leq 0.019$. In order to find the optimal value of $\beta$ within the above range, we adopt an grid search method with a step size of 0.002. The goal is to find the value of $\beta$ that minimizes perplexity of the sparse LLMs on the WikiText-2 (Merity et al., 2016) dataset. Since the reasonable range of $\beta$ is very small, this ensures that we can find the optimal $\beta$ very quickly. For example, for $0 < \beta \leq 0.019$, we only need to make 9 attempts. Even for the largest 70B model, the reasonable range of $\beta$ is $0 < \beta \leq 0.0075$, and only 3 attempts are required in this case. We present our ATP approach in Algorithm 1.

---

**Algorithm 1** Using ATP method to obtain sparse LLMs

**Input:** Dense LLMs $\mathcal{M}_{dense}$, average sparsity rate $S$
**Output:** Sparse LLMs $\mathcal{M}_{sparse}$
Use grid search to get $\beta$ in Eq. 10;
Determine layer-wise sparsity rate according to Eq. 10;
Combine with post-training sparsity method (*e.g.,*
  SpaeseGPT or Wanda) to obtain sparsity mask $\mathbf{W}$;
Apply $\mathbf{W}$ to $\mathcal{M}_{dense}$ yields $\mathcal{M}_{sparse}$.

---

Although the above method of determining the layer-wise sparsity rate of LLMs by a monotonically increasing arithmetic progression is very simple, our theoretical analysis in Sec. 3.5 fully proves its rationality, and we have fully demonstrated through a large number of experiments in Sec. 4 that our method can effectively improve the accuracy of existing post-training sparsity methods and significantly outperforms current layer-wise sparsity methods.

### 3.5. Analysis of the Proposed Determining Sparsity Method

We propose the following theorem to prove that the method for determining the layer-wise sparsity rate proposed in Eq. 10 is theoretically close to the optimal solution:

**Theorem 3.5.** *The total reconstruction error obtained from the monotonically increasing sparsity scheme proposed in Eq. 10 is strictly less than that obtained from any non-monotonically increasing sparsity scheme.*

The proof of the above theorem is detailed in Appendix C.

In addition, we compare the layer-wise sparsity rates determined by Eq. 10 with those obtained by Bayesian search in Sec. 4.6. The experimental results show that our method is

*Table 1.* Comparison of the average zero-shot accuracy across 7 tasks for 70% sparse LLMs obtained using various sparsity methods.

| Method | Layerwise sparsity | LLaMA | | | | LLaMA2 | | | LLaMA3 |
|---|---|---|---|---|---|---|---|---|---|
| | | 7B | 13B | 30B | 65B | 7B | 13B | 70B | 8B |
| Dense | - | 61.74 | 63.84 | 67.41 | 68.57 | 61.88 | 65.00 | 69.14 | 65.62 |
| Wanda | Uniform | 37.45 | 40.79 | 53.35 | 57.76 | 35.33 | 38.88 | 58.48 | 35.42 |
| | OWL | 44.19 | 48.18 | 55.88 | 59.79 | 41.75 | 47.19 | 59.26 | 35.42 |
| | DSA | 43.72 | 48.64 | 55.21 | 58.24 | 36.55 | 43.36 | 58.25 | 37.85 |
| | AlphaPruning | 44.99 | 49.81 | 56.67 | 61.05 | 43.37 | 49.11 | 59.44 | 35.61 |
| | **ATP** | **47.03** | **51.60** | **57.69** | **61.92** | **45.83** | **52.11** | **60.91** | **41.28** |
| SparseGPT | Uniform | 43.60 | 48.00 | 53.64 | 60.18 | 43.07 | 47.38 | 60.84 | 43.02 |
| | OWL | 46.57 | 50.01 | 55.73 | 59.47 | 46.55 | 49.90 | 60.83 | 46.06 |
| | AlphaPruning | 46.58 | 50.63 | 56.42 | 60.32 | 46.20 | 50.86 | 61.03 | 45.27 |
| | **ATP** | **47.37** | **52.12** | **57.55** | **61.31** | **48.63** | **52.46** | **62.25** | **47.79** |

close to the optimal solution obtained by Bayesian search. Moreover, since our method doesn't require a long time search, it is highly efficient compared to Bayesian search.

All in all, the above theoretical analysis and experimental validation demonstrate the superiority of our method, indicating that the sparsity allocation scheme proposed in Eq. 10 is near-optimal both theoretically and empirically.

## 4. Experiments

### 4.1. Experimental Setup

**Models.** We evaluate ATP across a diverse range of widely-used LLMs, including LLaMA-1 (7B, 13B, 30B and 65B) (Touvron et al., 2023a), LLaMA-2 (7B, 13B and 70B) (Touvron et al., 2023b), LLaMA-3-8B (Meta, 2024), LLaMA-3.1-8B (Meta, 2024), LLaMA-3.2-1B and 3B (Meta, 2024), OPT-13B (Zhang et al., 2022), Vicuna-13B (Chiang et al., 2023), Qwen2.5-7B (Yang et al., 2024), and Mistral-7B (Jiang et al., 2023) and Mixtral-8x7B (Jiang et al., 2024).

**Evaluation.** Our evaluation protocol aligns with established sparsification methods for LLMs (*e.g.,*, SparseGPT (Frantar & Alistarh, 2023) and Wanda (Sun et al., 2023)), encompassing both zero-shot learning and language modeling capabilities. Specifically, we assess the perplexity of the models on the validation set of WikiText-2 (Merity et al., 2016) and evaluate zero-shot performance on seven downstream tasks: BoolQ (Clark et al., 2019), ARC Easy and Challenge (Clark et al., 2018), HellaSwag (Zellers et al., 2019), WinoGrande (Sakaguchi et al., 2021), OpenbookQA (Mihaylov et al., 2018), and PIQA (Bisk et al., 2020). Additionally, we measure performance on arithmetic and knowledge reasoning benchmarks, including 8-shot accuracy on the GSM8K dataset (Cobbe et al., 2021) and 5-shot accuracy on the MMLU dataset (Hendrycks et al., 2020).

**Baselines.** We apply the layer-wise sparsity rates determined by ATP to several state-of-the-art post-training sparsification methods, including SparseGPT (Frantar & Alistarh, 2023), Wanda (Sun et al., 2023), DSnoT (Zhang et al., 2023), Pruner-zero (Dong et al., 2024), and ALPS (Meng et al., 2024). Furthermore, we compare ATP with recent methods for determining layer-wise sparsity rates in LLMs, such as OWL (Yin et al., 2023), AlphaPruning (Lu et al., 2024), DSA (Li et al., 2024b), and ALS (Li et al., 2024c).

**More Models, Evaluations and Baselines.** In Sec. F, we present additional experimental results, including the application of ATP to multimodal and vision models, integration with other compression techniques, LoRA fine-tuning and sparsity-preserving PEFT methods, and comparisons with an expanded set of layer-wise sparsity baselines.

**Implementation Details.** Our pruning implementation builds upon the methods used by SparseGPT and Wanda, with the primary modification being the integration of layer-wise sparsity rates generated by ATP.

### 4.2. Zero-shot Tasks

**Quantitative Evaluation.** We report the average performance of 70% sparse LLMs across seven zero-shot tasks in Table 1. The results demonstrate that our ATP method consistently improves accuracy compared to the uniform sparsity baseline and significantly outperforms other state-of-the-art layer-wise sparsity methods. For instance, with the LLaMA3-8B model pruned using the Wanda method, ATP achieves a 3.43% higher accuracy than the best-performing DSA method, highlighting the effectiveness and superiority of ATP in enhancing sparse LLM performance.

**Varying Sparsity Rates.** We also evaluate the performance of sparse LLMs under reduced sparsity constraints.

Specifically, Table 2 presents the zero-shot accuracy of LLMs pruned using the Wanda method at a 50% sparsity rate. Even under this lower sparsity setting, ATP demonstrates substantial improvements in accuracy across all models, maintaining its performance advantage over existing layer-wise sparsity methods. This indicates that ATP is robust and effective in optimizing LLM accuracy under varying sparsity rates.

*Table 2.* Average zero-shot accuracy of sparse LLaMA2-7B/13B models pruned using the Wanda method at 50% sparsity rate.

| Method | 1-7B | 1-13B | 2-7B | 2-13B |
|---|---|---|---|---|
| Dense | 66.09 | 68.18 | 66.69 | 69.25 |
| Uniform | 60.06 | 65.22 | 63.03 | 65.73 |
| OWL | 61.92 | 65.91 | 63.88 | 68.02 |
| DSA | 61.90 | 65.40 | 63.89 | 67.65 |
| ALS | 61.59 | 65.05 | 64.12 | 67.11 |
| AlphaPruning | 61.91 | 66.19 | 64.20 | 67.80 |
| **ATP** | **62.72** | **66.39** | **64.49** | **68.18** |

\* The zero-shot evaluation setting used here follows the configuration outlined in the ALS paper. For more details, refer to Sec. D.

### 4.3. Language Modeling

We present the perplexity of 50% to 80% sparse LLaMA-7B and LLaMA2-7B models pruned using the Wanda method on the WikiText-2 dataset in Table 3. The results show that our ATP method achieves lower perplexity compared to other layer-wise sparsity methods. Furthermore, this advantage becomes increasingly significant at higher sparsity rates.

*Table 3.* WikiText-2 perplexity of sparse LLaMA-7B/2-7B obtained by Wanda across varying sparsity rates.

| | LLaMA-7B | | | | LLaMA2-7B | | | |
|---|---|---|---|---|---|---|---|---|
| Method | 50% | 60% | 70% | 80% | 50% | 60% | 70% | 80% |
| Uniform | 7.26 | 10.63 | 84.52 | 5889.13 | 6.92 | 10.96 | 74.26 | 1980.85 |
| OWL | 7.22 | 9.35 | 24.56 | 1002.87 | 6.87 | 9.80 | 30.38 | 629.99 |
| DSA | 7.17 | 9.38 | 24.52 | 1232.88 | 7.05 | 10.40 | 63.71 | 1638.81 |
| AlphaPruning | 7.18 | 9.47 | 23.86 | 698.56 | 6.88 | 9.78 | 28.87 | 1672.49 |
| **ATP** | **7.05** | **9.25** | **20.16** | **176.80** | **6.82** | **9.15** | **22.16** | **425.12** |

### 4.4. More LLM Architectures

We validate the effectiveness of our ATP method on LLMs with a broader range of architectures. Specifically, we use ATP method to obtain 70% sparse models, including LLaMA3.1-8B, LLaMA3.2-1B/3B, OPT-13B, Vicuna-13B, Qwen2.5-7B, Mistral-7B, and Mixtral-8x7B. These models cover a wide range of parameter sizes, from 1B to 46.7B, including both LLaMA-like architectures and sparsely ac-

tivated Mixture of Experts (MoE) models. We report the experimental results in Table 4. The results demonstrate that our ATP method consistently enhances the performance of LLMs across various architectures, further demonstrating its generalizability across different model architectures.

*Table 4.* The performance improvement of ATP on sparse LLMs across a wider range of architectures.

| Method | Model | Perplexity ($\downarrow$) | Accuracy ($\uparrow$) |
|---|---|---|---|
| Dense | LLaMA3.1-8B | 6.18 | 65.93 |
| Wanda | LLaMA3.1-8B | 109.99 | 36.10 |
| **w. ATP** | **LLaMA3.1-8B** | **72.05** | **41.59** |
| Dense | LLaMA3.2-1B | 9.65 | 52.82 |
| SparseGPT | LLaMA3.2-1B | 129.24 | 37.26 |
| **w. ATP** | **LLaMA3.2-1B** | **94.78** | **38.86** |
| Dense | LLaMA3.2-3B | 7.73 | 60.42 |
| SparseGPT | LLaMA3.2-3B | 65.97 | 39.50 |
| **w. ATP** | **LLaMA3.2-3B** | **46.64** | **42.51** |
| Dense | OPT-13B | 10.13 | 55.22 |
| Wanda | OPT-13B | 73.70 | 41.51 |
| **w. ATP** | **OPT-13B** | **33.98** | **44.58** |
| Dense | Vicuna-13B | 5.94 | 65.53 |
| Wanda | Vicuna-13B | 44.89 | 42.06 |
| **w. ATP** | **Vicuna-13B** | **20.24** | **53.19** |
| Dense | Qwen2.5-7B | 6.77 | 65.36 |
| Wanda | Qwen2.5-7B | 75.16 | 41.10 |
| **w. ATP** | **Qwen2.5-7B** | **43.28** | **42.82** |
| Dense | Mistral-7B | 5.28 | 66.17 |
| Wanda | Mistral-7B | 57.44 | 37.62 |
| **w. ATP** | **Mistral-7B** | **29.19** | **42.76** |
| Dense | Mixtral-8x7B | 3.86 | 69.35 |
| Wanda | Mixtral-8x7B | 18.22 | 48.36 |
| **w. ATP** | **Mixtral-8x7B** | **14.30** | **50.67** |

### 4.5. More Results

We provide more experimental results in the appendix. Specifically, in Sec. F.1, we demonstrate the enhanced performance of our ATP method on arithmetic and knowledge reasoning tasks for sparse LLMs. Additionally, in Secs. F.2 and F.3, we present the performance gains of ATP on sparse multimodal and vision models, respectively. In Sec. F.4, we highlight the performance improvements when integrating ATP with other compression techniques, including N:M sparsity, structured pruning, and quantization. Furthermore, we compare ATP with additional layer-wise sparsity baselines (Sec. F.5), demonstrate its performance enhancements across various post-training sparsity methods (Sec. F.6), and showcase its effectiveness when combined with LoRA fine-tuning for sparse LLMs (Sec. F.7). Furthermore, we show the zero-shot accuracy of LLMs at 60% sparsity in Sec. F.8.

## 4.6. Ablation Study

**Searching Step.** In Sec. 3.4, we perform a grid search with a step size of 0.002 to determine the optimal value of $\beta$. Here, we analyze the impact of different step sizes on the search results. As shown in Table 5, searches conducted with larger step sizes yield inferior results compared to those with a step size of 0.002. This is because larger step sizes fail to sufficiently explore the possible optimal values of $\beta$. Conversely, further reducing the step size for a more fine-grained search shows limited improvement in perplexity. Therefore, to balance both accuracy and efficiency, we adopt a grid search with a step size of 0.002.

*Table 5.* The impact of different step sizes on search results.

| Step size | 0.0005 | 0.001 | 0.002 | 0.004 | 0.008 |
|---|---|---|---|---|---|
| Perplexity ($\downarrow$) | 22.14 | 22.16 | 22.16 | 23.09 | 23.09 |
| Search Time (min) | 76 | 38 | 18 | 8 | 4 |

**Comparison with Bayesian Search.** We compare the ATP method with layer-wise sparsity rates obtained through Bayesian search. Specifically, we use Bayesian search to determine the sparsity rates for the 50%, 60%, and 70% sparse LLaMA2-7B model. The search is conducted using the Optuna hyperparameter optimization framework (Akiba et al., 2019), performing 1000 iterations to optimize the sparsity rates across all 32 layers of LLaMA2-7B. The sparsity rate for each layer is constrained between 0 and 1, while ensuring that the average sparsity matches the target rates. We integrate Bayesian search with the Wanda method, with the objective of minimizing model perplexity on the WikiText-2 dataset. The comparison of perplexity and zero-shot accuracy between sparse LLMs obtained using ATP and Bayesian search is presented in Table 6. The results indicate that the performance of our ATP method is comparable to the optimal solution obtained through Bayesian search, demonstrating that the layer-wise sparsity rates determined by our method are experimentally close to the optimal values identified by the search approach. However, Bayesian search requires approximately 33 hours to complete on a single NVIDIA A100 80GB GPU, whereas ATP only takes 18 minutes, demonstrating significantly higher efficiency.

*Table 6.* Comparison of ATP and Bayesian search.

| Method | Sparsity | Perplexity ($\downarrow$) | Accuracy ($\uparrow$) |
|---|---|---|---|
| Bayesian Search | 50% | 6.81 | 59.67 |
| **ATP** | **50**% | **6.82** | **59.63** |
| Bayesian Search | 60% | 9.20 | 54.73 |
| **ATP** | **60**% | **9.15** | **54.79** |
| Bayesian Search | 70% | 22.10 | 45.90 |
| **ATP** | **70**% | **22.16** | **45.83** |

**Inference Speedup.** We evaluate the acceleration performance of the sparse LLaMA2-7B model, with results summarized in Table 7. The end-to-end token generation time was measured using the DeepSparse (NeuralMagic, 2021) inference engine on an Intel Xeon Silver 4314 CPU and the nm-vllm (NeuralMagic, 2024) inference engine on an NVIDIA RTX 4090 GPU. Our method achieves significant speedups, ranging from $1.79\times$ to $2.63\times$ on the CPU and $1.71\times$ to $2.23\times$ on the GPU, compared to the dense model, at sparsity rates between 50% and 70%. In addition, there is a discussion about which devices can be used with the DeepSparse and nm-vllm inference engines to deploy sparse LLMs for inference acceleration, please refer to Sec. E.

*Table 7.* End-to-end inference acceleration of sparse LLaMA2-7B on CPU and GPU.

| Device | Sparsity | Dense | 50% | 60% | 70% |
|---|---|---|---|---|---|
| CPU | Throughput (tokens/s) $\uparrow$ | 3.40 | **6.10** | **7.39** | **8.95** |
|  | **Speedup** $\uparrow$ | 1.00$\times$ | **1.79$\times$** | **2.17$\times$** | **2.63$\times$** |
| GPU | Throughput (tokens/s) $\uparrow$ | 57.29 | **97.92** | **111.86** | **127.67** |
|  | **Speedup** $\uparrow$ | 1.00$\times$ | **1.71$\times$** | **1.95$\times$** | **2.23$\times$** |

**More Ablation Study.** We provide more ablation results in the appendix. Specifically, in Sec. G.1, we demonstrate the computational efficiency of our ATP method. In Sec. G.2, we analyze the layer-wise sparsity distribution generated by ATP at different sparsity rates. In addition, we compare the sparsity rate distributions produced by ATP and other layer-wise sparsity methods in Sec. G.3. In Sec. G.4, we explore the impact of different $\beta$ settings on the perplexity of sparse LLMs. Furthermore, we demonstrate the robustness of ATP under different random seeds in Sec. G.5.

## 5. Conclusion

In this paper, we propose a theoretically grounded method for determining layer-wise sparsity rates in LLMs, effectively addressing the challenge of **"reconstruction error explosion"**. Our approach utilizes an arithmetic progression to streamline sparsity allocation, reducing the complexity to a single hyperparameter while achieving near-optimal performance with minimal tuning. We have demonstrated through theoretical analysis and experimental validation that our sparsity allocation scheme is close to the optimal solution. Extensive experiments demonstrate the effectiveness of our method, yielding significant improvements in model perplexity, accuracy, and inference speed. Furthermore, our approach exhibits strong generalization across diverse architectures and modalities, establishing it as a versatile and robust solution for optimizing compressed models.

## Acknowledgments

This work was supported by the National Science Fund for Distinguished Young Scholars (No.62025603), the National Natural Science Foundation of China (No. U21B2037, No. U22B2051, No. U23A20383, No. 62176222, No. 62176223, No. 62176226, No. 62072386, No. 62072387, No. 62072389, No. 62002305 and No. 62272401), and the Natural Science Foundation of Fujian Province of China (No. 2021J06003, No.2022J06001).

## Impact Statement

This paper presents work whose goal is to advance the field of Machine Learning. There are many potential societal consequences of our work, none which we feel must be specifically highlighted here.

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

# A. Limitation and Future Work

In this paper, we derive that layer-wise sparsity rates should gradually increase based on the reconstruction error analysis and determine these rates using a monotonically increasing arithmetic progression. Extensive experiments validate the effectiveness of our method. However, the arithmetic progression configuration may not be optimal, and we plan to explore more diverse sparsity rate schemes in the future. Additionally, while our method significantly enhances the accuracy of existing post-training sparsity techniques for LLMs, there remains a performance gap compared to lossless sparsification, particularly under high sparsity conditions. In future work, we aim to develop more advanced post-training sparsity methods to further improve the accuracy of sparse LLMs.

# B. Proof of Lemma 3.3

*Proof of Lemma 3.3.* Let $A \in \mathbb{R}^{m \times n}$ and $B \in \mathbb{R}^{n \times p}$ be any matrices. Consider the singular value decomposition (SVD) of $A$:

$$A = U\Sigma V^T \tag{11}$$

where:

- $U \in \mathbb{R}^{m \times m}$ is an orthogonal matrix ($U^T U = I$),

- $V \in \mathbb{R}^{n \times n}$ is an orthogonal matrix ($V^T V = I$), and

- $\Sigma \in \mathbb{R}^{m \times n}$ is a diagonal matrix with non-negative singular values $\sigma_1 \geq \sigma_2 \geq \cdots \geq \sigma_{\min(m,n)} \geq 0$ on the diagonal.

Substituting the SVD of $A$ into the expression for $AB$, we have:

$$AB = U\Sigma V^T B \tag{12}$$

Therefore,

$$\|AB\|_F^2 = \left\| U\Sigma V^T B \right\|_F^2 \tag{13}$$

Since $U$ is an orthogonal matrix, the Frobenius norm is invariant under orthogonal transformations. Specifically:

$$\|UX\|_F = \|X\|_F \quad \forall X \tag{14}$$

Applying this property:

$$\|AB\|_F^2 = \left\| \Sigma V^T B \right\|_F^2 \tag{15}$$

The matrix $\Sigma$ is diagonal with singular values $\sigma_i$ on the diagonal. Let $\sigma_{\min} = \sigma_{\min}(A)$ denote the smallest singular value of $A$. Then, for any matrix $X$, we have:

$$\Sigma X \geq \sigma_{\min} X \tag{16}$$

in the sense that each singular value scales the corresponding component of $X$.

Therefore, applying this to our case:

$$\left\| \Sigma V^T B \right\|_F^2 \geq \sigma_{\min}^2 \left\| V^T B \right\|_F^2 \tag{17}$$

This inequality holds because scaling each component by at least $\sigma_{\min}$ results in the squared norm being scaled by at least $\sigma_{\min}^2$.

Similarly to $U$, the orthogonal matrix $V$ preserves the Frobenius norm:

$$\left\| V^T X \right\|_F = \|X\|_F \quad \forall X \tag{18}$$

Applying this property:

$$\left\| V^T B \right\|_F = \|B\|_F \tag{19}$$

Substituting back, we obtain:

$$\|\boldsymbol{AB}\|_F^2 = \left\|\boldsymbol{\Sigma V}^T\boldsymbol{B}\right\|_F^2 \geq \sigma_{\min}^2 \left\|\boldsymbol{V}^T\boldsymbol{B}\right\|_F^2 = \sigma_{\min}^2 \|\boldsymbol{B}\|_F^2 \tag{20}$$

Thus, we have shown that:

$$\|\boldsymbol{AB}\|_F^2 \geq \sigma_{\min}^2(\boldsymbol{A}) \|\boldsymbol{B}\|_F^2 \tag{21}$$

$\square$

## C. Proof of Theorem 3.5

*Proof of Theorem 3.5.* Recall that each layer $i \in \{1, 2, \ldots, L\}$ in an LLM has a sparsity rate $s_i \in [0, 1]$, where $s_i$ indicates the fraction of zero entries in the weight matrix of layer $i$. Let $\mathcal{L}_i$ denote the reconstruction error of layer $i$, and let $\mathcal{L} = \sum_{i=1}^{L} \mathcal{L}_i$ be the total reconstruction error of the entire LLM.

To prove Theorem 3.5, we will make the following assumptions based on the preceding sections:

1. According to Theorem 3.1, The reconstruction error for each layer $i$, denoted as $\mathcal{L}_i$, is an increasing function of the sparsity rate $s_i$, denote as $f(s_i)$.

2. According to Theorem 3.2, the reconstruction error propagates through layers such that the increase of reconstruction error of $i$-th layer will lead to the increase of reconstruction error of $(i + 1)$-th layer. Therefore we assume $\mathcal{L}_{i+1} = c\mathcal{L}_i + f(s_i)$, where $c > 1$. The above formula indicates that the reconstruction error of a layer not only accumulates the reconstruction errors from previous layers but also that the current sparse layer contributes to the reconstruction error.

The monotonically increasing sparsity scheme given by Eq. 10 (in Sec. 3.4) implies that $s_1 < s_2 < \cdots < s_L$, let us denote such a scheme as

$$\mathbf{s}^{\uparrow} = (s_1^{\uparrow}, s_2^{\uparrow}, \ldots, s_L^{\uparrow}), \tag{22}$$

where $s_1^{\uparrow} < s_2^{\uparrow} < \cdots < s_L^{\uparrow}$.

Consider a different non-monotonically increasing sparsity scheme:

$$\mathbf{s}^{\diamond} = (s_1^{\diamond}, s_2^{\diamond}, \ldots, s_L^{\diamond}), \tag{23}$$

which is *not* monotonically increasing. That is, there exists at least one index $k$ such that $s_k^{\diamond} > s_{k+1}^{\diamond}$

Moreover, suppose both $\mathbf{s}^{\uparrow}$ and $\mathbf{s}^{\diamond}$ satisfy the constraint that their average sparsities match the same overall budget. Formally,

$$\frac{1}{L}\sum_{i=1}^{L} s_i^{\uparrow} = \frac{1}{L}\sum_{i=1}^{L} s_i^{\diamond} = S, \tag{24}$$

and each $\mathbf{s}^{\uparrow}$ and $\mathbf{s}^{\diamond}$ results in a total reconstruction error

$$\mathcal{L}^{\uparrow} = \sum_{i=1}^{L}\mathcal{L}_i^{\uparrow}, \quad \mathcal{L}^{\diamond} = \sum_{i=1}^{L}\mathcal{L}_i^{\diamond}. \tag{25}$$

We will show that for any non-monotonic sparsity vector $\mathbf{s}^{\diamond}$, we can *redundantly reorder* it layer by layer to get a monotonically increasing vector $\mathbf{s}^{\uparrow}$ of the same average sparsity, such that the overall reconstruction error $\mathcal{L}^{\uparrow}$ is *strictly smaller* than $\mathcal{L}^{\diamond}$. Ultimately, we will deduce

$$\mathcal{L}^{\uparrow} < \mathcal{L}^{\diamond}. \tag{26}$$

Let us focus on any adjacent pair $(s_k^{\diamond}, s_{k+1}^{\diamond})$ where $s_k^{\diamond} > s_{k+1}^{\diamond}$. Define $s_k^{*} = s_{k+1}^{\diamond}$ and $s_{k+1}^{*} = s_k^{\diamond}$, effectively *swapping* these two sparsities. We then compare the total contribution of layers $k$ and $k + 1$ to the overall reconstruction error, first in the non-monotonic case and then in the swapped case.

We restrict our attention to layers $k$ and $k+1$:

$$\text{Layer } k \quad \Rightarrow \quad \mathcal{L}_k^\diamond = f(s_k^\diamond), \qquad \text{Layer } k+1 \quad \Rightarrow \quad \mathcal{L}_{k+1}^\diamond = c\,\mathcal{L}_k^\diamond + f(s_{k+1}^\diamond). \tag{27}$$

Strictly speaking, $\mathcal{L}_{k+1}^\diamond$ depends on partial errors from layer $k$ and its own sparsity $s_{k+1}^\diamond$. Intuitively, a higher error $\mathcal{L}_k^\diamond$ "propagates" or "magnifies" into layer $k+1$. We keep the rest of the layer-wise sparsities fixed outside of these two positions to isolate their effect.

Now, consider *swapping* the two sparsities: $(s_k^\diamond,\, s_{k+1}^\diamond) \mapsto (s_k^*,\, s_{k+1}^*) = (s_{k+1}^\diamond,\, s_k^\diamond)$. Let

$$\mathcal{L}_k^* = f(s_k^*) = f(s_{k+1}^\diamond), \quad \mathcal{L}_{k+1}^* = c\,\mathcal{L}_k^* + f(s_{k+1}^*) = c\,f(s_{k+1}^\diamond) + f(s_k^\diamond). \tag{28}$$

Therefore, the total for these two layers is

$$\mathcal{L}_{\text{swapping}} = \mathcal{L}_k^* + \mathcal{L}_{k+1}^* \;=\; f(s_{k+1}^\diamond) + \Big(c\,f(s_{k+1}^\diamond) + f(s_k^\diamond)\Big) \;=\; (1+c)\,f(s_{k+1}^\diamond) \;+\; f(s_k^\diamond). \tag{29}$$

Meanwhile, the original total is

$$\mathcal{L}_{\text{original}} = \mathcal{L}_k^\diamond + \mathcal{L}_{k+1}^\diamond \;=\; f(s_k^\diamond) \;+\; \Big(c\,f(s_k^\diamond) + f(s_{k+1}^\diamond)\Big) \;=\; (1+c)\,f(s_k^\diamond) \;+\; f(s_{k+1}^\diamond). \tag{30}$$

The difference between the two is

$$\begin{aligned}
\mathcal{L}_{\text{original}} - \mathcal{L}_{\text{swapping}} &= \Big[(1+c)\,f(s_k^\diamond) + f(s_{k+1}^\diamond)\Big] \;-\; \Big[(1+c)\,f(s_{k+1}^\diamond) + f(s_k^\diamond)\Big] \\
&= (1+c)\big[f(s_k^\diamond) - f(s_{k+1}^\diamond)\big] \;-\; \big[f(s_k^\diamond) - f(s_{k+1}^\diamond)\big] \\
&= c\,\big[f(s_k^\diamond) - f(s_{k+1}^\diamond)\big].
\end{aligned} \tag{31}$$

Since $c > 1$ and $f(s_k^\diamond) > f(s_{k+1}^\diamond)$, it follows that $c\,\big[f(s_k^\diamond) - f(s_{k+1}^\diamond)\big] \;>\; 0$.

Hence we obtain

$$\mathcal{L}_{\text{original}} \;>\; \mathcal{L}_{\text{swapping}}. \tag{32}$$

This shows that *locally swapping* a pair $(s_k^\diamond, s_{k+1}^\diamond)$ with $s_k^\diamond > s_{k+1}^\diamond$ *reduces* the total reconstruction error contributed by layers $k$ and $k+1$. Globally across all $L$ layers, repeating such a swap for each adjacent pair that breaks the monotonicity moves $\mathbf{s}^\diamond$ to a strictly monotonically increasing sequence of sparsities.

By iterating this argument over each adjacent pair that fails monotonicity, we reorder $\mathbf{s}^\diamond$ into $\mathbf{s}^\uparrow$. Since every swap strictly reduces the local error, the resulting *monotonically increasing* scheme $\mathbf{s}^\uparrow$ has an overall reconstruction error:

$$\mathcal{L}^\uparrow \;=\; \sum_{i=1}^{L} \mathcal{L}_i^\uparrow \;<\; \sum_{i=1}^{L} \mathcal{L}_i^\diamond \;=\; \mathcal{L}^\diamond. \tag{33}$$

Therefore, the total reconstruction error obtained from the monotonically increasing sparsity scheme proposed in Eq. 10 is strictly less than that obtained from any non-monotonically increasing sparsity scheme, under the same average sparsity constraint.

$$\square$$

## D. Zero-shot Evaluation Setting

We use the lm-eval-harness framework (Gao et al., 2021) to evaluate the zero-shot performance of LLMs. By default, we follow the settings used in Wanda (Sun et al., 2023) and AlphaPruning (Lu et al., 2024), reporting the "acc" metric across all datasets. However, lm-eval-harness provides multiple metrics depending on the dataset, including both "acc" and "acc_norm". In contrast, ALS (Li et al., 2024c) employs different metrics for different datasets. The evaluation metrics are summarized in Table 8.

Table 8. Evaluation metrics for different datasets

| Dataset | Metric |
|---------|--------|
| boolq | acc |
| hellaswag | acc_norm |
| arc_easy | acc |
| piqa | acc |
| arc_challenge | acc_norm |
| winogrande | acc |
| openbookqa | acc_norm |

## E. Sparse Inference Engine Supported Devices

Due to the sparsity of weights in unstructured pruning, we must use a specific sparse inference engine to accelerate inference. We use DeepSparse and nm-vllm to accelerate inference on general deployment environments, including CPUs and GPUs. For CPUs, DeepSparse supports architectures including: x86 AVX2, AVX-512, AVX-512 VNNI, and ARM v8.2+, which covers most Intel, AMD, and Apple M-series CPUs. For GPUs, as long as the device supports CUDA, inference acceleration can be achieved using the nm-vllm. Similarly, if CUDA is supported for other deployment environments, nm-vllm can also be used to achieve acceleration.

## F. More Results

### F.1. Arithmetic and Knowledge Reasoning Tasks

We further evaluate the performance improvements of our ATP method on arithmetic and knowledge reasoning tasks for sparse models. Specifically, We evaluate the 8-shot accuracy of 40% sparse models on the GSM8K dataset and the 5-shot accuracy of 60% sparse models on the MMLU dataset. The results are presented in Table 9. Our ATP method significantly improves the accuracy of sparse models on both tasks, further demonstrating the generalization and effectiveness of our approach.

Table 9. Accuracy of sparse LLaMA2 on the GSM8K and MMLU datasets.

| Method | GSM8K | | MMLU | |
|--------|-------|-----|------|-----|
| | 7B | 13B | 7B | 13B |
| Dense | 14.60 | 24.56 | 45.90 | 55.70 |
| SparseGPT | 11.07 | 16.91 | 29.40 | 39.60 |
| **SparseGPT w.ATP** | **12.01** | **19.02** | **32.40** | **47.70** |
| Wanda | 8.72 | 17.51 | 28.90 | 34.80 |
| **Wanda w.ATP** | **9.93** | **19.56** | **32.80** | **46.20** |

### F.2. Multimodal Tasks

We demonstrate the applicability of our method to multimodal models by integrating it with Wanda to sparsify the Vicuna-7B model (Chiang et al., 2023) in LLaVA-1.5 (Liu et al., 2024b). We only sparsify the Vicuna model within it. Similar to LLaMA, we determine the sparsity rate for each layer and sparsify linear layers. The performance of the sparse model was evaluated on various visual question-answering and reasoning benchmarks, including VQA (Singh et al., 2019), VQAv2 (Goyal et al., 2017), and SQA (Lu et al., 2022). We compared our method with magnitude-based pruning, SparseGPT, Wanda, and the DSA method combined with Wanda, under a 50% sparsity rate. The results in Table 10 show that our method outperforms the magnitude-based, SparseGPT, and Wanda approaches and demonstrates superiority over the DSA method.

### F.3. Vision Models

We compare our ATP method with other approaches for determining layer-wise sparsity rates on vision models. Experiments are conducted on both CNN and Transformer architectures, including ConvNeXt (Liu et al., 2022b), ViT (Dosovitskiy et al.,

*Table 10.* The performance improvement of the ATP method on sparse multimodal models.

| LLaVA-1.5 | VQA | VQAv2 | SQA |
|---|---|---|---|
| Dense | 58.20 | 78.50 | 66.80 |
| Magnitude | 38.39 | 63.50 | 31.24 |
| SparseGPT | 53.69 | 75.86 | 63.92 |
| Wanda | 53.05 | 75.72 | 63.99 |
| Wanda w. DSA | 54.36 | 76.08 | 65.57 |
| **Wanda w. ATP** | **55.21** | **76.92** | **66.01** |

2021), and DeiT (Touvron et al., 2021), and we report the Top-1 accuracy on the ImageNet-1K dataset (Deng et al., 2009). Sparse models are obtained using the Wanda (Sun et al., 2023) method, and ATP is compared against the following baselines: uniform sparsity rate, OWL (Yin et al., 2023), and AlphaPruning (Lu et al., 2024). Following Wanda, we only sparsify the linear layers within each block of the ConvNeXt, and we use our ATP method to determine the sparsity rate for each block. For ViT, we use ATP to determine the sparsity rate for each layer. Each layer contains modules such as attention and MLP. We use Wanda to sparsify linear layers. The results for ConvNeXt are presented in Table 11. These results demonstrate that our method is effective for determining layer-wise sparsity rates in vision models and outperforms existing methods. This further confirms the broad applicability of our monotonically increasing sparsity scheme in improving the accuracy of diverse sparse models.

*Table 11.* Comparison of ImageNet-1K accuracy across different layer-wise sparsity methods on the ConvNeXt-Base model.

| Method | **50**% | **60**% | **70**% | **80**% |
|---|---|---|---|---|
| Wanda | 82.72 | 80.55 | 68.18 | 6.44 |
| w. OWL | 82.76 | 80.53 | 68.28 | 6.32 |
| w. AlphaPruning | 82.76 | 80.89 | 74.24 | 42.35 |
| **w. ATP** | **82.85** | **81.10** | **75.58** | **56.81** |

We also present the accuracy results for sparse ViT and DeiT models in Table 12. Additionally, we compare our method with several widely-used non-uniform layer-wise sparsity strategies in computer vision, including ERK (Mocanu et al., 2018), Global (Frankle & Carbin, 2018), and LAMP (Lee et al., 2020), as well as AlphaPruning (Lu et al., 2024). The results indicate that our method outperforms all the aforementioned baselines.

*Table 12.* Comparison of ImageNet-1K accuracy across different layer-wise sparsity methods on the ViT-B and DeiT-B models.

| Method | ViT-B 16/224 | | | DeiT-B 16/224 | | |
|---|---|---|---|---|---|---|
| | 40% | 50% | 60% | 40% | 50% | 60% |
| Uniform | 70.87 | 59.46 | 29.97 | 80.08 | 76.37 | 61.72 |
| ERK (Mocanu et al., 2018) | 70.89 | 60.49 | 33.15 | 80.05 | 76.22 | 63.49 |
| Global (Frankle & Carbin, 2018) | 66.81 | 45.75 | 8.09 | 79.94 | 75.09 | 57.01 |
| LAMP (Lee et al., 2020) | 69.45 | 57.51 | 26.99 | 80.19 | 76.35 | 63.32 |
| AlphaPruning (Lu et al., 2024) | 71.58 | 64.29 | 44.21 | 80.21 | 77.11 | 64.56 |
| **ATP** | **72.03** | **65.46** | **47.74** | **80.50** | **78.02** | **69.73** |

## F.4. Integrate with other Compression Technologies

To demonstrate the generalization ability of our method for determining layer-wise sparsity rates, we combine ATP with other compression techniques, including N:M sparsity, structured pruning, and mixed-precision quantization. For N:M sparsity, we follow the mixed N:8 settings (Sun et al., 2021), using ATP to determine the value of $N$ for each layer while maintaining average sparsity at 2:8, 3:8 and 4:8. In terms of structured pruning, we integrated ATP with LLM-Pruner (Ma et al., 2023a), where ATP determines the layer-wise sparsity rates, and LLM-Pruner applies pruning accordingly. For mixed-precision quantization, we combine ATP with the LIMPQ method (Tang et al., 2022) to determine the quantization bits for each layer. We apply these compression techniques to the LLaMA-7B model and report the perplexity of the compressed models on the WikiText-2 validation set. The experimental results are presented in Table 13. It shows that ATP

significantly enhances the performance of various compression methods and outperforms both the OWL and AlphaPruning approaches.

*Table 13.* Experimental results of combining ATP with other compression technologies.

| Method | 4:8 | 3:8 | 2:8 |
|---|---|---|---|
| Wanda | 8.57 | 42.56 | 2962.00 |
| OWL | 8.55 | 22.77 | 331.37 |
| AlphaPruning | 8.55 | 21.49 | 585.81 |
| **ATP** | **8.15** | **16.08** | **265.96** |
| **Method** | **20**% | **40**% | **60**% |
| LLM-Pruner | 16.95 | 30.38 | 90.02 |
| OWL | 18.57 | 28.65 | 76.99 |
| AlphaPruning | 16.78 | 29.11 | 71.21 |
| **ATP** | **15.51** | **27.40** | **64.25** |
| **Method** | **Mixed 3/4 Bit** | **Mixed 2/3/4 Bit** | **Mixed 2/4 Bit** |
| Random | 12.04 | 11455.54 | 14817.12 |
| $L_1$ norm | 14.61 | 13959.42 | 33679.21 |
| OWL | 9.54 | 311.95 | 8429.39 |
| AlphaPruning | 9.01 | 261.39 | 7630.14 |
| **ATP** | **8.52** | **198.65** | **5321.35** |

## F.5. More Layer-wise Sparsity Baselines

We compare our method with additional approaches for determining layer-wise sparsity rates, including Uniform (Zhu & Gupta, 2017), Global (Frankle & Carbin, 2018), ER (Mocanu et al., 2018), ER-Plus (Liu et al., 2022a), and LAMP (Lee et al., 2020). These methods are combined with the Wanda pruning approach to obtain sparse LLaMA-7B models. The experimental results are presented in Table 14. It shows that across sparsity rates ranging from 50% to 80%, the perplexity of models pruned using the ATP method is consistently lower than that of other baselines, further demonstrating the superiority of our approach.

*Table 14.* Comparison of WikiText-2 perplexity across various sparsity rates and methods.

| Method/Perplexity ($\downarrow$) | 50% | 60% | 70% | 80% |
|---|---|---|---|---|
| Global | 14848 | 17765 | 5147 | 39918.56 |
| LAMP | 7.57 | 12.86 | 185.52 | 15647.87 |
| LAMP (per-block) | 7.25 | 10.95 | 98.77 | 7318.08 |
| ER | 7.80 | 12.41 | 119.66 | 6263.79 |
| ER-Plus | 8.00 | 14.04 | 229.17 | 6013.91 |
| Uniform | 7.26 | 10.63 | 84.52 | 5889.13 |
| **ATP** | **7.05** | **9.25** | **20.16** | **176.80** |

## F.6. Integrate with More Post-training Sparsity Methods

We have demonstrated that our method improves performance over Wanda and SparseGPT methods. Notably, ATP can be integrated with any post-training sparsity method to further enhance their effectiveness. In this section, we showcase the performance improvements achieved by combining ATP with other post-training sparsity methods. Specifically, we apply ATP to DSnoT (Zhang et al., 2023), Pruner-Zero (Dong et al., 2024), and ALPS (Meng et al., 2024) to obtain a 70% sparse LLaMA2-7B model. The perplexity and zero-shot accuracy results of the sparse models are presented in Table 15. The results indicate that ATP significantly enhances the performance of DSnoT, Pruner-Zero, and ALPS methods.

## F.7. Integrate with LoRA Fine-tuning

We further demonstrate the effectiveness of LoRA fine-tuning (Hu et al., 2021) in narrowing the performance gap between highly sparse LLMs and dense models. Specifically, we obtain a 70% sparse LLaMA2-7B model using the Wanda method and fine-tune it on 10,000 samples from the Alpaca-GPT4 (Peng et al., 2023) dataset. We compare the results against models

*Table 15.* The performance improvement of ATP on DSnoT, Pruner-zero and ALPS methods.

| Method | Perplexity (↓) | Accuracy (↑) |
|---|---|---|
| Dense | 5.12 | 61.88 |
| DSnoT | 77.83 | 35.11 |
| **w. ATP** | **24.31** | **44.51** |
| Pruner-Zero | 103.15 | 34.78 |
| **w. ATP** | **48.82** | **44.77** |
| ALPS | 19.31 | 46.75 |
| **w. ATP** | **17.99** | **49.82** |

sparsified using uniform sparsity, OWL, and AlphaPruning methods. The results in Table 16 show that LoRA fine-tuning significantly improves the accuracy of the sparse model, further reducing the gap with the dense model. Additionally, the sparse model obtained through the ATP method achieves higher accuracy, and this advantage is retained even after LoRA fine-tuning.

*Table 16.* Comparison of the perplexity and zero-shot accuracy of the 70% sparse LLaMA2-7B obtained by LoRA fine-tuning.

| Method | Fine-tuning | Perplexity (↓) | Accuracy (↑) |
|---|---|---|---|
| Dense | N.A. | 5.12 | 61.88 |
| Uniform | N.A. | 74.26 | 35.33 |
| Uniform | LoRA | 13.36 | 47.10 |
| OWL | N.A. | 30.38 | 41.75 |
| OWL | LoRA | 12.89 | 50.26 |
| DSA | N.A. | 63.71 | 36.55 |
| DSA | LoRA | 13.19 | 49.49 |
| AlphaPruning | N.A. | 28.87 | 43.37 |
| AlphaPruning | LoRA | 12.76 | 50.37 |
| **ATP** | **N.A.** | **22.16** | **45.83** |
| **ATP** | **LoRA** | **12.28** | **50.79** |

## F.8. Zero-shot Accuracy at 60% Sparsity

We present the zero-shot accuracy results of LLaMA3-8B at 60% sparsity in Table 17. The sparse model is obtained using the Wanda method, and we compare ATP with other layer-wise sparsity methods. We can observe that ATP significantly improves the accuracy of Wanda, increasing the average zero-shot accuracy by 4.13% and narrowing the performance gap between sparse and dense model. Furthermore, ATP outperforms OWL, DSA, and AlphaPruning, with 1.05% higher accuracy than the best-performing AlphaPruning.

*Table 17.* Zero-shot accuracy results of sparse LLaMA3-8B at 60% sparsity obtained using the Wanda method.

| Method | HellaSwag | Winogrande | BoolQ | OBQA | PIQA | ARC-e | ARC-c | Mean |
|---|---|---|---|---|---|---|---|---|
| Dense | 60.19 | 72.77 | 81.35 | 34.80 | 79.71 | 80.09 | 50.43 | 65.62 |
| Uniform | 37.92 | 59.90 | 68.16 | 19.80 | 67.55 | 59.63 | 27.43 | 48.63 |
| OWL | 40.71 | 62.90 | 70.34 | 22.90 | 69.80 | 62.28 | 31.39 | 51.47 |
| DSA | 40.16 | 63.01 | 70.09 | 22.80 | 69.05 | 62.20 | 30.51 | 51.12 |
| AlphaPruning | 41.63 | 64.51 | 71.25 | 22.60 | 68.61 | 62.88 | 30.46 | 51.71 |
| **ATP** | **41.91** | **65.49** | **71.68** | **23.70** | **70.94** | **63.62** | **31.96** | **52.76** |

## G. More Ablation Study

### G.1. Computational Efficiency.

In Table 18, we report the time required to determine the layer-wise sparsity rates for 70% sparse LLMs using our ATP method. The measurements were conducted on NVIDIA A100 80GB GPUs. The results show that only a few searches are

needed to obtain the sparsity rates due to the narrow range of reasonable values for the hyperparameter $\beta$. Furthermore, as the number of model parameters increases, this range narrows even further, enabling the sparsity rates to be determined with fewer searches. For example, for the largest 70B model, only three searches are necessary. This demonstrates that our method is highly computationally efficient.

*Table 18.* Computational efficiency of our ATP method (in minutes).

| Model | 7B | 8B | 13B | 30B | 65B | 70B |
|---|---|---|---|---|---|---|
| Wanda | 2.0 | 2.2 | 3.8 | 7.8 | 13.5 | 14.1 |
| w. ATP | 2.0×9 | 2.2×9 | 3.8×7 | 7.8×5 | 13.5×3 | 14.1×3 |
| SparseGPT | 6.6 | 7.3 | 10.0 | 35.0 | 60.0 | 67.5 |
| w. ATP | 6.6×9 | 7.3×9 | 7.3×7 | 10.0×5 | 60.0×3 | 67.5×3 |

## G.2. Analyze the Layer-wise Sparsity Distribution.

We analyze the sparsity rate distribution across different average sparsity levels in Figure 2. Our findings indicate that at lower average sparsity rates, the differences in sparsity rates across layers are minimal. In contrast, these differences become more pronounced at higher average sparsity rates.

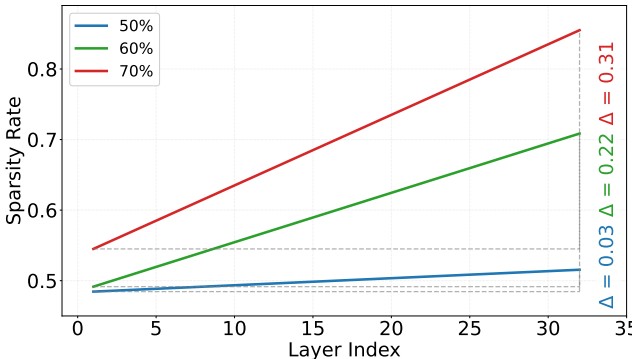

*Figure 2.* Comparison of layer-wise sparsity rate distributions at different average sparsity levels.

## G.3. Comparison of Sparsity Rates Distribution with other Methods

Figure 3 illustrates the sparsity rate distributions obtained from various layer-wise sparsity methods. We observe that the sparsity rates generally follow an increasing pattern from low to high, further validating the rationale behind our method.

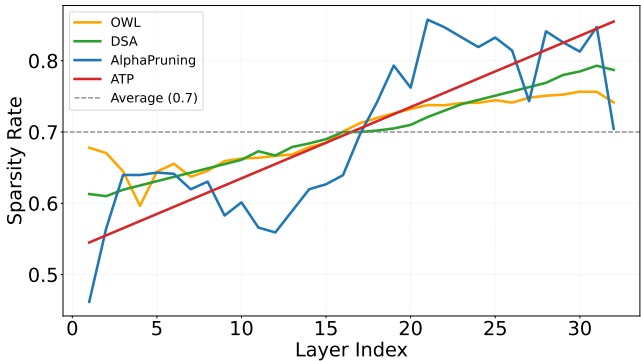

*Figure 3.* Comparison of layer-wise sparsity rate distribution with other methods.

### G.4. Different $\beta$ Settings.

Figure 4 shows the impact of different $\beta$ settings on the perplexity of the 70% sparse LLaMA2-7B model obtained using the Wanda method. We observe that as $\beta$ increases, the perplexity initially decreases and then rises. Furthermore, the model's perplexity under various $\beta$ settings remains lower than that of the uniform sparsity rate scheme.

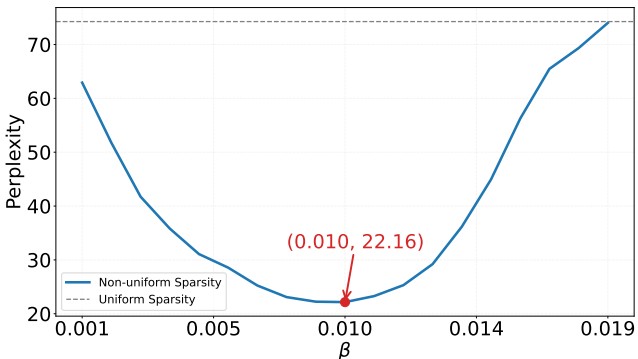

*Figure 4.* Impact of different $\beta$ settings on the perplexity of the 70% sparse LLaMA2-7B.

### G.5. Robustness of ATP under Different Random Seeds

Following OWL (Yin et al., 2023), DSA (Li et al., 2024b) and AlphaPruning (Lu et al., 2024), all experimental results are conducted under a single fixed random seed. We also report WikiText2 perplexity of 70% sparse LLaMA2-7B obtained by Wanda across five random seeds and different calibration sets in Table 19. The variance across random seeds is very low, suggesting the robustness of ATP.

*Table 19.* The WikiText2 perplexity of 70% sparse LLaMA2-7B obtained by Wanda across five random seeds.

| Method | Perplexity |
|---|---|
| Dense | 5.12 ($\pm$) 0.00 |
| Uniform | 74.26 ($\pm$) 0.10 |
| OWL | 30.38 ($\pm$) 0.09 |
| DSA | 63.71 ($\pm$) 0.08 |
| AlphaPruning | 28.87 ($\pm$) 0.06 |
| **ATP** | **22.16 ($\pm$) 0.05** |

## H. Comparison with NEURONAL

The previous work NEURONAL (Cunegatti et al., 2024) also proposed adopting a monotonically increasing sparsity schedule for LLMs, but there are significant differences between our ATP and NEURONAL:

1. **Significant differences in algorithms for determining monotonically increasing sparsity schedules for LLMs.**

   Although both ATP and NEURONAL propose using monotonically increasing arithmetic progression to allocate sparsity rates for each block of LLMs, there are significant differences in the algorithms by which ATP and NEURONAL derive the final arithmetic sequences.

   ATP calculates the reasonable range of values for the common difference $\beta$ of the arithmetic progression, and then searches for the optimal value of $\beta$ within this reasonable range using a grid search with a step size of 0.002. The goal of search is to find the value of $\beta$ that minimizes perplexity of the sparse LLMs on the WikiText-2 dataset.

   In contrast, NEURONAL defines a sparsity window, $[s - \lambda, s + \lambda]$, in advance. Given an average sparsity rate of s, NEURONAL searches for the optimal $\lambda$ value within a predefined range of $[0.01, 0.02, 0.03, 0.05, 0.06, 0.07, 0.08, 0.09, 0.10, 0.12, 0.15, 0.20, 0.25]$. The optimal $\lambda$ value is determined by maximizing the alignment between the activation

values of the sparse and dense models. Due to the differences in these algorithms, the final sparsity allocation schemes obtained by ATP and NEURONAL are different.

2. **Significant differences between the two sparsity methods beyond the monotonically increasing sparsity schedule.**

   ATP focuses on deriving that existing LLM sparsity methods have a reconstruction error explosion problem, thus proposing the use of a monotonically increasing arithmetic sequence to determine the sparsity rates for LLMs to alleviate the above problem. Our method provides theoretical insights into the effectiveness and rationality of monotonically increasing arithmetic sequence sparsity schemes. We only determined non-uniform sparsity rates for each block of LLMs, with linear layers within blocks using uniform sparsity rates.

   In contrast, NEURONAL proposes utilizing functional information from dense pre-trained models, i.e., their activations, to obtain sparse models that maximize alignment of activations with the corresponding dense model. NEURONAL adaptively selects the best hyperparameters for block sparsity rates and row-wise sparsity rates based on the model and desired sparsity level, to maximize neuron alignment between activations.

   Therefore, ATP and NEURONAL also have significant differences at the more macro algorithmic level of sparsity algorithms.

## I. Detailed Results for Zero-shot Tasks

In this section, we provide a detailed presentation of the performance of each zero-shot task introduced in Sec. 4.

*Table 20.* Zero-shot accuracy results of LLaMA-V1 at 70% sparsity.

| Model | Method | HellaSwag | Winogrande | BoolQ | OBQA | PIQA | ARC-e | ARC-c | Mean |
|---|---|---|---|---|---|---|---|---|---|
| | Dense | 56.92 | 69.93 | 75.05 | 34.40 | 78.67 | 75.34 | 41.89 | 61.74 |
| | SparseGPT | 34.58 | 56.43 | 64.80 | 16.80 | 64.25 | 45.24 | 23.12 | 43.60 |
| | w. OWL | 37.08 | 62.35 | 66.15 | 19.80 | 66.16 | 48.40 | 26.02 | 46.57 |
| | w. AlphaPruning | 37.81 | 64.01 | **68.26** | 18.80 | 63.60 | 46.17 | 27.39 | 46.58 |
| LLaMA-7B | **w. ATP** | **37.88** | **64.22** | 66.51 | **20.00** | **66.17** | **49.37** | **27.47** | **47.37** |
| | Wanda | 28.86 | 52.80 | 59.69 | 14.20 | 57.56 | 31.27 | 17.75 | 37.45 |
| | w. OWL | 34.89 | 58.64 | 62.63 | 17.60 | 64.30 | 46.97 | 24.32 | 44.19 |
| | w. DSA | 34.68 | 59.67 | 62.69 | 15.40 | 64.09 | 45.16 | 24.40 | 43.72 |
| | w. AlphaPruning | 36.26 | **62.35** | **66.12** | 17.20 | 63.93 | 43.90 | 25.17 | 44.99 |
| | **w. ATP** | **37.44** | 61.43 | **65.79** | **20.80** | **67.46** | **50.63** | **25.68** | **47.03** |
| | Dense | 59.94 | 72.77 | 77.89 | 33.20 | 79.16 | 77.40 | 46.50 | 63.84 |
| | SparseGPT | 37.51 | 63.30 | 68.78 | 20.80 | 67.63 | 52.78 | 25.17 | 48.00 |
| | w. OWL | 40.36 | 66.22 | 66.82 | 20.80 | 69.86 | 57.37 | 28.67 | 50.01 |
| | w. AlphaPruning | 42.43 | 67.80 | 67.58 | 22.00 | 70.01 | 55.47 | 29.10 | 50.63 |
| LLaMA-13B | **w. ATP** | **43.31** | **67.92** | **69.14** | **23.40** | **70.89** | **59.97** | **30.20** | **52.12** |
| | Wanda | 31.06 | 54.38 | 61.59 | 16.20 | 62.68 | 42.05 | 17.58 | 40.79 |
| | w. OWL | 38.57 | 63.46 | 62.81 | 20.40 | 68.61 | 57.07 | 26.37 | 48.18 |
| | w. DSA | 38.84 | 62.04 | 63.03 | 23.20 | 68.98 | 58.29 | 26.10 | 48.64 |
| | w. AlphaPruning | 41.04 | 64.96 | 64.83 | 21.20 | 69.01 | 59.39 | 28.24 | 49.81 |
| | **w. ATP** | **43.11** | **64.98** | **66.64** | **25.80** | **69.80** | **59.97** | **30.89** | **51.60** |
| | Dense | 63.35 | 75.69 | 82.69 | 36.00 | 81.01 | 80.30 | 52.82 | 67.41 |
| | SparseGPT | 44.56 | 69.30 | 65.35 | 25.80 | 72.42 | 65.78 | 32.25 | 53.64 |
| | w. OWL | 46.96 | 72.20 | 67.95 | 26.80 | 73.56 | 67.13 | 35.49 | 55.73 |
| | w. AlphaPruning | 47.49 | 72.30 | 68.96 | 30.00 | 73.50 | 67.97 | 34.73 | 56.42 |
| LLaMA-30B | **w. ATP** | **48.77** | **72.83** | **69.28** | **30.60** | **73.94** | **68.81** | **36.68** | **57.55** |
| | Wanda | 44.23 | 67.01 | 66.70 | 26.40 | 72.03 | 64.86 | 32.25 | 53.35 |
| | w. OWL | 47.69 | 69.77 | 65.02 | 29.20 | 73.88 | 68.98 | 36.62 | 55.88 |
| | w. DSA | 45.62 | 67.25 | 73.15 | 27.80 | 72.36 | 68.13 | 32.17 | 55.21 |
| | w. AlphaPruning | 49.40 | 71.27 | 63.82 | 31.00 | 73.50 | 69.40 | 38.31 | 56.67 |
| | **w. ATP** | **50.27** | **71.37** | 65.35 | **31.40** | **75.41** | **70.62** | **39.42** | **57.69** |
| | Dense | 64.53 | 77.27 | 84.89 | 38.00 | 81.23 | 81.36 | 52.73 | 68.57 |
| | SparseGPT | 49.98 | 74.74 | 81.03 | 29.20 | 74.76 | 72.05 | 39.51 | 60.18 |
| | w. OWL | 51.25 | 74.98 | 81.30 | 27.00 | 75.68 | 68.35 | 37.71 | 59.47 |
| | w. AlphaPruning | 52.40 | 74.59 | 83.80 | 28.42 | 75.68 | 69.30 | 38.05 | 60.32 |
| LLaMA-65B | **w. ATP** | **53.10** | **75.69** | 83.73 | **28.45** | **76.06** | **71.46** | **40.69** | **61.31** |
| | Wanda | 46.29 | 70.79 | 77.37 | 27.20 | 74.21 | 70.66 | 37.79 | 57.76 |
| | w. OWL | 50.90 | 74.11 | 78.50 | 30.60 | 75.68 | 70.70 | 38.05 | 59.79 |
| | w. DSA | 48.60 | 73.08 | 71.77 | 28.80 | 75.35 | 71.88 | 38.22 | 58.24 |
| | w. AlphaPruning | 52.55 | 75.06 | 81.85 | 30.40 | 75.84 | 71.60 | 40.02 | 61.05 |
| | **w. ATP** | **52.92** | **75.70** | **82.73** | **31.80** | **75.98** | **73.53** | **40.78** | **61.92** |

*Table 21.* Zero-shot accuracy results of LLaMA-V2/V3 at 70% sparsity.

| Model | Method | HellaSwag | Winogrande | BoolQ | OBQA | PIQA | ARC-e | ARC-c | Mean |
|---|---|---|---|---|---|---|---|---|---|
| LLaMA2-7B | Dense | 57.17 | 68.90 | 77.74 | 31.40 | 78.07 | 76.39 | 43.52 | 61.88 |
| | SparseGPT | 33.08 | 58.41 | 64.89 | 17.40 | 62.46 | 43.22 | 22.01 | 43.07 |
| | w. OWL | 36.57 | 63.06 | 66.94 | 21.60 | 63.89 | 49.33 | 24.49 | 46.55 |
| | w. AlphaPruning | 36.83 | 62.19 | 65.93 | 19.80 | 64.58 | 49.62 | 24.43 | 46.20 |
| | **w. ATP** | **38.94** | **63.54** | **69.54** | **21.70** | **68.12** | **52.74** | **25.85** | **48.63** |
| | Wanda | 27.92 | 49.33 | 52.87 | 12.60 | 55.33 | 30.60 | 18.69 | 35.33 |
| | w. OWL | 31.83 | 55.96 | 62.11 | 16.80 | 61.70 | 43.52 | 20.31 | 41.75 |
| | w. DSA | 28.54 | 50.36 | 57.46 | 12.00 | 57.13 | 32.95 | 17.41 | 36.55 |
| | w. AlphaPruning | 34.56 | 60.85 | 62.23 | 18.00 | 62.40 | 43.27 | 22.27 | 43.37 |
| | **w. ATP** | **36.08** | **61.01** | **62.39** | **20.40** | **66.81** | **50.76** | **23.38** | **45.83** |
| LLaMA2-13B | Dense | 60.06 | 72.22 | 80.52 | 35.20 | 79.11 | 79.42 | 48.46 | 65.00 |
| | SparseGPT | 36.90 | 61.64 | 66.02 | 21.00 | 67.57 | 52.61 | 25.94 | 47.38 |
| | w. OWL | 39.31 | 65.75 | 68.04 | 22.80 | 67.89 | 57.70 | 27.82 | 49.90 |
| | w. AlphaPruning | 41.26 | 68.03 | 68.13 | 24.00 | 68.28 | 57.15 | 29.18 | 50.86 |
| | **w. ATP** | **42.81** | **68.09** | **72.91** | **24.40** | **69.74** | **58.25** | **31.06** | **52.46** |
| | Wanda | 29.60 | 51.70 | 62.32 | 13.60 | 58.65 | 37.21 | 19.11 | 38.88 |
| | w. OWL | 36.31 | 60.46 | 63.46 | 21.80 | 67.77 | 55.64 | 24.91 | 47.19 |
| | w. DSA | 32.83 | 55.01 | 62.41 | 17.80 | 63.71 | 49.87 | 21.92 | 43.36 |
| | w. AlphaPruning | 40.28 | 67.32 | 62.57 | 21.60 | 68.17 | 54.46 | 29.35 | 49.11 |
| | **w. ATP** | **41.44** | **67.50** | **74.71** | **24.60** | **68.25** | **57.49** | **30.80** | **52.11** |
| LLaMA2-70B | Dense | 66.10 | 78.06 | 83.40 | 37.20 | 82.21 | 82.55 | 54.44 | 69.14 |
| | SparseGPT | 50.98 | **75.45** | 80.06 | 30.00 | 75.24 | 73.57 | 40.61 | 60.84 |
| | w. OWL | 51.95 | 74.98 | 79.25 | 30.40 | 75.68 | 73.00 | 40.53 | 60.83 |
| | w. AlphaPruning | 51.90 | 75.06 | 80.40 | 29.20 | 75.68 | 74.10 | 40.87 | 61.03 |
| | **w. ATP** | **53.44** | **76.56** | **80.42** | **31.00** | **76.71** | **75.38** | **42.23** | **62.25** |
| | Wanda | 48.16 | 73.88 | 74.46 | 27.00 | 74.86 | 72.69 | 38.31 | 58.48 |
| | w. OWL | 50.20 | 74.02 | 75.01 | 28.60 | 75.68 | 73.02 | 38.30 | 59.26 |
| | w. DSA | 48.60 | 73.08 | 71.77 | 28.80 | 75.35 | 71.88 | 38.22 | 58.25 |
| | w. AlphaPruning | 51.40 | 74.74 | 73.65 | 29.60 | 75.84 | 72.70 | 38.13 | 59.44 |
| | **w. ATP** | **51.84** | **75.53** | **78.65** | **29.70** | **76.66** | **74.83** | **39.16** | **60.91** |
| LLaMA3-8B | Dense | 60.19 | 72.77 | 81.35 | 34.80 | 79.71 | 80.09 | 50.43 | 65.62 |
| | SparseGPT | 34.26 | 56.75 | 66.51 | 16.80 | 63.28 | 42.09 | 21.42 | 43.02 |
| | w. OWL | 36.78 | 58.96 | 69.54 | 18.20 | 65.45 | 49.46 | 24.06 | 46.06 |
| | w. AlphaPruning | 35.54 | 61.56 | 71.02 | 17.00 | 63.92 | 46.17 | 21.67 | 45.27 |
| | **w. ATP** | **38.19** | **63.22** | **71.12** | **19.00** | **66.59** | **50.20** | **26.19** | **47.79** |
| | Wanda | 27.36 | 49.96 | 53.33 | 12.00 | 56.04 | 31.86 | 17.41 | 35.42 |
| | w. OWL | 28.43 | 50.43 | 61.74 | 13.00 | 57.99 | 35.82 | 17.58 | 37.85 |
| | w. DSA | 27.51 | 48.46 | 54.16 | 11.80 | 56.63 | 33.03 | 17.66 | 35.61 |
| | w. AlphaPruning | 27.82 | 51.85 | 56.42 | 13.40 | 56.47 | 34.97 | 17.32 | 36.89 |
| | **w. ATP** | **31.46** | **54.93** | **61.79** | **16.40** | **62.18** | **41.79** | **20.39** | **41.28** |

*Table 22.* Zero-shot accuracy results of LLaMA-V1/V2 at 50% sparsity.

| Model | Method | HellaSwag | Winogrande | BoolQ | OBQA | PIQA | ARC-e | ARC-c | Mean |
|---|---|---|---|---|---|---|---|---|---|
| LLaMA-7B | Dense | 76.18 | 70.09 | 75.11 | 44.40 | 79.16 | 72.98 | 44.71 | 66.09 |
| | Wanda | 68.92 | 66.38 | 70.70 | 39.00 | 74.76 | 61.74 | 38.91 | 60.06 |
| | w. OWL | 70.06 | 66.85 | 71.44 | 39.80 | 75.52 | 69.44 | 40.36 | 61.92 |
| | w. DSA | 69.51 | 67.95 | 71.31 | 39.40 | 75.46 | 69.65 | 40.01 | 61.90 |
| | w. ALS | 69.59 | 66.30 | 73.70 | 38.60 | **77.26** | 65.66 | 40.02 | 61.59 |
| | w. AlphaPruning | 69.60 | 67.64 | 73.33 | 39.20 | 75.57 | 69.65 | 38.39 | 61.91 |
| | **w. ATP** | **70.18** | **68.06** | **73.92** | **39.90** | 76.51 | **69.74** | **40.70** | **62.72** |
| LLaMA-13B | Dense | 79.06 | 72.77 | 77.89 | 44.80 | 80.14 | 74.79 | 47.78 | 68.18 |
| | Wanda | 74.13 | 71.51 | 75.96 | 43.60 | 77.91 | 69.65 | 43.77 | 65.22 |
| | w. OWL | 74.82 | 72.53 | 76.39 | 43.60 | 77.37 | 73.32 | 43.34 | 65.91 |
| | w. DSA | 74.02 | 70.79 | 76.05 | 43.20 | 76.98 | 72.60 | 44.19 | 65.40 |
| | w. ALS | 74.34 | 71.35 | 75.17 | 43.00 | 77.37 | 69.70 | 44.45 | 65.05 |
| | w. AlphaPruning | 74.58 | 72.77 | 76.63 | 44.00 | 76.82 | 74.20 | 44.36 | 66.19 |
| | **w. ATP** | **74.86** | **72.78** | **76.47** | **44.20** | **77.46** | **74.36** | **44.60** | **66.39** |
| LLaMA2-7B | Dense | 75.98 | 69.06 | 77.74 | 44.20 | 79.11 | 74.49 | 46.25 | 66.69 |
| | Wanda | 68.76 | 67.32 | 75.78 | 41.40 | 76.99 | 69.23 | 41.72 | 63.03 |
| | w. OWL | 70.79 | 67.32 | 75.96 | 42.80 | 76.33 | 72.01 | 41.97 | 63.88 |
| | w. DSA | 70.90 | 66.45 | 76.42 | 43.00 | 76.22 | 71.42 | 42.83 | 63.89 |
| | w. ALS | 70.75 | 67.80 | 75.47 | **44.80** | 77.10 | 69.61 | 42.32 | 64.12 |
| | w. AlphaPruning | 70.89 | 67.48 | 76.63 | 43.00 | 76.33 | 72.22 | 42.83 | 64.20 |
| | **w. ATP** | **70.99** | **67.84** | **76.73** | 43.70 | **76.44** | **72.90** | **42.85** | **64.49** |
| LLaMA2-13B | Dense | 79.39 | 72.38 | 80.58 | 45.20 | 80.52 | 77.53 | 49.15 | 69.25 |
| | Wanda | 75.02 | 69.39 | 80.34 | 44.10 | 78.13 | 70.37 | 42.76 | 65.73 |
| | w. OWL | 76.11 | 71.19 | 81.65 | 45.40 | 78.67 | 76.85 | 46.24 | 68.02 |
| | w. DSA | 75.86 | 71.03 | 80.83 | 45.00 | 78.62 | 76.22 | 45.99 | 67.65 |
| | w. ALS | 75.67 | 72.06 | 81.35 | **45.80** | 78.51 | 70.33 | 46.08 | 67.11 |
| | w. AlphaPruning | 76.19 | 71.58 | 80.97 | 45.00 | 78.34 | 76.38 | 46.16 | 67.80 |
| | **w. ATP** | **76.26** | **72.09** | **81.66** | 45.20 | **78.74** | **76.86** | **46.42** | **68.18** |

*Table 23.* Zero-shot accuracy results of more LLM architectures at 70% sparsity.

| Model | Method | HellaSwag | Winogrande | BoolQ | OBQA | PIQA | ARC-e | ARC-c | Mean |
|---|---|---|---|---|---|---|---|---|---|
| LLaMA3.1-8B | Dense | 59.98 | 73.32 | 82.05 | 33.20 | 79.98 | 81.57 | 51.45 | 65.93 |
| | Wanda | 27.43 | 48.70 | 57.71 | 13.60 | 55.01 | 31.86 | 18.43 | 36.10 |
| | **w. ATP** | **31.81** | **55.49** | **62.08** | **15.60** | **62.68** | **42.63** | **20.82** | **41.59** |
| OPT-13B | Dense | 52.43 | 65.04 | 65.93 | 27.20 | 75.84 | 67.13 | 32.94 | 55.22 |
| | Wanda | 34.36 | 55.09 | 55.02 | 15.60 | 62.89 | 43.73 | 23.89 | 41.51 |
| | **w. ATP** | **36.47** | **58.09** | **62.17** | **18.20** | **65.56** | **46.63** | **24.91** | **44.58** |
| Vicuna-13B | Dense | 59.64 | 71.59 | 85.26 | 36.80 | 79.00 | 78.66 | 47.78 | 65.53 |
| | Wanda | 31.84 | 54.70 | 62.78 | 16.40 | 61.75 | 44.87 | 22.10 | 42.06 |
| | **w. ATP** | **41.47** | **63.85** | **76.70** | **25.00** | **69.04** | **61.95** | **34.30** | **53.19** |
| Qwen2.5-7B | Dense | 60.01 | 72.85 | 84.65 | 33.20 | 78.73 | 80.43 | 47.70 | 65.36 |
| | Wanda | 30.68 | 51.93 | 61.96 | 15.20 | 61.59 | 46.34 | 20.05 | 41.10 |
| | **w. ATP** | **32.84** | **56.59** | **62.14** | **16.40** | **63.44** | **46.42** | **21.93** | **43.82** |
| Mistral-7B | Dense | 61.21 | 73.88 | 83.64 | 32.60 | 80.58 | 80.85 | 50.43 | 66.17 |
| | Wanda | 28.86 | 51.07 | 60.03 | 12.60 | 57.56 | 34.60 | 18.69 | 37.62 |
| | **w. ATP** | **34.44** | **58.96** | **62.20** | **15.60** | **63.93** | **42.68** | **21.50** | **42.76** |

*Table 24.* Zero-shot accuracy results of more post-training sparsity methods at 70% sparsity.

| Model | Method | HellaSwag | Winogrande | BoolQ | OBQA | PIQA | ARC-e | ARC-c | Mean |
|---|---|---|---|---|---|---|---|---|---|
| LLaMA2-7B | Dense | 57.17 | 68.90 | 77.74 | 31.40 | 78.07 | 76.39 | 43.52 | 61.88 |
| | DSnoT | 27.80 | 51.78 | 49.66 | 12.60 | 55.66 | 30.89 | 17.41 | 35.11 |
| | **w. ATP** | **34.98** | **58.01** | **62.23** | **18.20** | **65.78** | **50.42** | **21.93** | **44.51** |
| | Pruner-Zero | 27.56 | 50.99 | 41.93 | 13.00 | 56.90 | 34.47 | 18.60 | 34.78 |
| | **w. ATP** | **35.97** | **57.93** | **64.46** | **18.60** | **64.15** | **48.27** | **23.98** | **44.77** |
| | ALPS | 38.35 | 61.96 | 64.59 | 22.20 | 66.82 | 48.37 | 24.95 | 46.75 |
| | **w. ATP** | **41.58** | **64.25** | **64.78** | **24.20** | **68.34** | **56.65** | **28.92** | **49.82** |

