# OpenReview forum: "Determining Layer-wise Sparsity for Large Language Models Through a Theoretical Perspective"
_ICML.cc/2025/Conference — ICML 2025 spotlightposter_

### Official Review · Reviewer_Jh19 · 2025-03-10

**Overall Recommendation:** 4

**Summary:**

This paper determines the sparsity rate of each layer for LLMs from a theoretical perspective, proposes that there is a "reconstruction error explosion" problem in sparse LLMs, and proposes to use a monotonically increasing arithmetic progression to determine the sparsity rate of each layer, thereby alleviating the "reconstruction error explosion" problem. Experimental results on different models and datasets show the effectiveness of this method and that it outperforms existing methods.

## update after rebuttal

I am satisfied with the authors' response, including clarifications and additional experiments. I will keep my positive rating, and I hope the responses could be incorporated into the next revision.

**Claims And Evidence:**

The author claims that there exists a "reconstruction error explosion" problem in sparse LLMs which leads to severe accuracy degradation. This claim is supported by specific observations, and in Figure 1, we can see the trend of reconstruction error explosion. The author also analyzes the reasons for the "reconstruction error explosion" from a theoretical perspective, and I think there is no issue with the author's claim. Additionally, the author claims that using a monotonically increasing arithmetic progression can alleviate the above problem, thereby improving the accuracy of sparse LLMs. Extensive experimental validations support this claim made by the author.

**Essential References Not Discussed:**

From the best of my knowledge, no essential references are not discussed. All the authors discuss are the latest sparsity methods for LLMs. The layer-wise sparsity methods that are most relevant to this paper have been discussed by the authors.

**Experimental Designs Or Analyses:**

The authors' experimental design follows established protocols for LLM sparsification methods, including Wanda, SparseGPT, and other layer-wise sparsity approaches for comparison. The paper's experiments encompass various LLM architectures and sizes across numerous datasets, demonstrating results at different sparsity rates. I find the experimental design to be well-reasoned and effective in evaluating the proposed method's efficacy.

**Methods And Evaluation Criteria:**

The layer-wise sparsity method based on monotonically increasing arithmetic progression proposed in this paper makes sense, as it effectively improves the accuracy of sparse LLMs, outperforming existing layer-wise sparsity methods.

**Other Comments Or Suggestions:**

All the lines in the left picture of Figure 1 are close together and need to be zoomed in to be seen clearly. Can the author improve the clarity of this picture?

**Other Strengths And Weaknesses:**

Strengths:
1. The paper demonstrates good organization, with a methodically presented approach supported by numerous figures, tables, and reproducible code.
2. The idea of a monotonically increasing arithmetic progression to determine the layer-wise sparsity rate is interesting and novel, and the authors reveal the rationality of this sparsity rate scheme from the perspective of "reconstruction error explosion".
3. The authors provide comprehensive experimental results on various tasks and models. Compared with the existing layer-wise sparsity baselines, APT outperforms the existing baselines. The efficiency of models make them good to deploy in real-world.

Weaknesses:
For larger parameter LLMs, the improvements from the ATP method appear to be less substantial than those observed in smaller LLMs.

**Questions For Authors:**

1. What are the practical advantages of LLMs sparsity as adopted in this paper compared to other acceleration techniques (such as quantization)?
2. The authors show the zero-shot accuracy of LLMs at 50% and 70% sparsity. I would like to know what the zero-shot accuracy of the ATP method would be under the 60% sparsity setting?

**Relation To Broader Scientific Literature:**

The author divides the previous layer-wise sparsity methods for LLMs into two categories, including metric based methods (OWL, AlphaPruning and ALS) and search based methods (DSA). The former requires complex calculations and lacks theoretical guarantees of optimality, while the latter demands time-consuming search processes spanning days and the search effect is heavily dependent on human expertise. Therefore, the author proposes to derive the layer-wise sparsity rate of LLMs from a theoretical perspective, so that only a simple monotonically increasing arithmetic progression is needed, and the author guarantees from a theoretical perspective that this scheme is near optimal. I think the analysis of the issues existing in the existing layer-wise sparsity methods is reasonable, and the motivation behind determining the layer-wise sparsity rate from a theoretical perspective is great.

**Theoretical Claims:**

I have examined the author's proof of the theorem and did not find any issues. The author's proof mainly involves expanding and transforming the reconstruction error based on the Frobenius norm, thereby proving that increasing the sparsity rate in earlier layers leads to an increase in reconstruction error in later layers, which consequently results in the "reconstruction error explosion".

---

> ### Author Rebuttal · Authors · 2025-03-31
>
> **Thanks for your careful review and comments!**
>
> > Weaknesses (Smaller improvements for larger models.)
>
> We have observed that larger models show less performance improvement with ATP. This observation aligns with common patterns in model pruning and compression, where the returns tend to diminish as model size increases. However, our ATP still demonstrates consistent improvements across different model sizes, especially notable in larger models:
>
> 1. **Significance of Relative Gains**
>
> While absolute improvements may appear smaller for larger models, the relative gains remain significant. For example, for LLaMA-65B at 70% sparsity with Wanda, ATP reduces the zero-shot accuracy loss from 10.81% to 6.65%, representing a 38.48% relative improvement. However, the improvement of AlphaPruning is only 30.71%. Such gains can translate to meaningful performance enhancements in real-world applications.
>
> 2. **Inherent Resilience of Larger Models**
>
> Larger models naturally exhibit greater resistance to pruning. As shown in Table 1, 70% sparse LLaMA2-70B with SparseGPT experiences an zero-shot accuracy loss of only 8.30%, compared to 18.81% for LLaMA2-7B, leaving less room for further improvement. Nevertheless, our method further reduces the loss of  LLaMA2-70B to 6.89%. This observation aligns with the findings of Li et al. [1].
>
> 3. **Alignment with Theoretical Insights**
>
> The behavior aligns with recent theoretical insights. Liu et al. [2] demonstrated that larger models can maintain performance even under random pruning, supporting the idea of their inherent pruning resistance.
>
> In summary, our ATP method remains effective across all model sizes and still achieves considerable performance improvements in larger models, providing critical stability and performance benefits.
>
> [1] Li et al. Train big, then compress: Rethinking model size for efficient training and inference of transformers.
>
> [2] Liu et al. The unreasonable effectiveness of random pruning: Return of the most naive baseline for sparse training.
> > Other Comments Or Suggestions (Figure 1 needs to be enlarged.)
>
> Thank you for your suggestion. We will enlarge Figure 1 in our final version to make it clearer for viewing.
> > Question 1 (What are the advantages of sparsity compared to quantization?)
>
> Network sparsity and quantization are both effective methods for improving inference speed and reducing memory footprints, with both techniques offering significant acceleration benefits across different hardware platforms (such as CPU and GPU). However, comparisons between these two approaches in terms of efficiency may vary.
>
> Recent research suggests that network sparsity can slightly outperform quantization in inference speedup. For example, SqueezeLLM [3] demonstrates that quantizing LLaMA2-7B to 3 bit achieves a 2.1$\times$ speedup on GPU. Meanwhile, the latest sparse CPU and GPU kernels (DeepSparse and nm-vllm) provide better support for deploying sparse LLMs, achieving 2.63$\times$ CPU inference acceleration and 2.23$\times$ GPU inference acceleration for 70% sparse LLaMA2-7B, as shown in Table 6 of our paper. Additionally, compared to quantization, sparsity can better maintain and recover performance through fine-tuning, as described in [4].
>
> Pruning and quantization are compatible and complementary approaches, combining these methods can further enhance efficiency. Table 3 in [4] shows that combining network sparsity with quantization (INT8) can achieve up to 9.08$\times$ acceleration, significantly higher than using either method alone.
>
> [3] SqueezeLLM: Dense-and-Sparse Quantization.
>
> [4] Sparse Fine-tuning for Inference Acceleration of Large Language Models.
> > Question 2 (Experimental results with 60% sparsity.)
>
> We present the zero-shot accuracy results of Llama-3-8B at 60% sparsity below. The sparse model is obtained using the Wanda method, and we compare ATP with other layer-wise sparsity methods.
>
> |Method|HellaSwag|Winogrande|BoolQ|OBQA|PIQA|ARC-e|ARC-c|Mean|
> |-|-|-|-|-|-|-|-|-|
> |Dense|60.19|72.77|81.35|34.80|79.71|80.09|50.43|65.62|
> |Uniform|37.92|59.90|68.16|19.80|67.55|59.63|27.43|48.63|
> |OWL|40.71|62.90|70.34|22.90|69.80|62.28|31.39|51.47|
> |DSA|40.16|63.01|70.09|22.80|69.05|62.20|30.51|51.12|
> |AlphaPruning|41.63|64.51|71.25|22.60|68.61|62.88|30.46|51.71|
> |**ATP**|**41.91**|**65.49**|**71.68**|**23.70**|**70.94**|**63.62**|**31.96**|**52.76**|
>
> We observe that:
>
> 1. ATP significantly improves the accuracy of Wanda, increasing the average zero-shot accuracy by 4.13% and narrowing the performance gap between sparse and dense model.
>
> 2. ATP outperforms OWL, DSA, and AlphaPruning, with 1.05% higher accuracy than the best-performing AlphaPruning.
>
> In conclusion, our ATP method demonstrates consistent improvements across various sparsity levels and significantly outperforms existing layer-wise sparsity methods.
>
> **We will incorporate the above responses into the final version. We hope that our response has addressed your concerns. Thank you!**

---

### Official Review · Reviewer_JETH · 2025-03-11

**Overall Recommendation:** 4

**Summary:**

This paper identifies the issue of "reconstruction error explosion" in existing LLMs sparsification methods. Through theoretical analysis, it derives a layer-wise sparsity allocation method based on a monotonically increasing arithmetic progression to alleviate the above issue. Both theoretical analysis and experimental results indicate that the above sparsity allocation scheme is near optimal, and significantly improves the performance of sparse LLMs with various architectures, outperforming the existing layer-wise sparsity methods.

**Claims And Evidence:**

1. The author claims that the existing post-training sparsification methods for LLMs suffer from the issue of "reconstruction error explosion". The author supports the above claim through proof of Theorems 3.1-3.4. Moreover, the author has depicted the "explosive" trend of the reconstruction error of sparse LLMs increasing with the layer number in Figure 1, which also supports the above claim.

2. The author proposes that the issue of "reconstruction error explosion" can be alleviated through a layer-wise sparsity rate scheme based on a monotonically increasing arithmetic progression. The author has proven the effectiveness of the above method through extensive experiments. Moreover, through the proof of Theorem 3.5 and the experiment comparing with Bayesian search, the author supports the claim that the above scheme is near optimal both theoretically and experimentally.

3. All in all, the author has effectively supported the proposed claims through theoretical analysis and experimental verification.

**Essential References Not Discussed:**

No. The author compares the existing methods for determining the layer-wise sparsity rates of LLMs, including OWL, ALS, DSA, and AlphaPruning. These are the layer-wise sparsity methods customized for LLMs that were published at ICML or NeurIPS prior to the submission date of this paper. As far as I know, there were no other layer-wise sparsity rate methods specifically designed for LLMs before the submission date. Additionally, the author demonstrates the improvement effects of these layer-wise sparsity methods on the Wanda and SparseGPT methods, which are the latest and widely used sparsity methods for LLMs. In the appendix, the author also compares the methods for determining the layer-wise sparsity rates of CNNs and ViT models. Overall, the author has compared and discussed with many relevant and essential works.

**Experimental Designs Or Analyses:**

Yes. The author has conducted sufficient experiments, comparing with the SOTA baselines. Experiments have been carried out on different LLMs, many tasks have been evaluated, and experiments have also been conducted on multimodal models and vision models. The author has also demonstrated the effects of combining with different compression techniques. In addition, the author has analyzed the ablation experiments of different hyperparameters, the computational efficiency of the proposed method, as well as the distribution of the layer-wise sparsity rate. Both the experimental design and analysis are reasonable and effective.

**Methods And Evaluation Criteria:**

The author discovers the issue of "reconstruction error explosion" existing in the sparsity methods of LLMs. This refers to the cumulative effect of reconstruction errors. Throughout the entire sparsification process, the errors from the early layers propagate and amplify in the subsequent layers, leading to a significant increase in the overall reconstruction error and a substantial decline in the model's performance. This explains well that why the existing sparse LLMs have relatively low accuracy. Moreover, the author proposes to alleviate this issue by using a layer-wise sparsity scheme based monotonically increasing arithmetic progression, which is intuitive and reasonable. The author's theoretical analysis and experimental verification both show that this layer-wise sparsity scheme makes sense.

**Other Comments Or Suggestions:**

Typos:
Line 833: Table 7. Evaluation Metrics for Different Datasets. -> Table 7. Evaluation metrics for different datasets.

**Other Strengths And Weaknesses:**

Strengths:

1. Originality: The layer-wise sparsity rate scheme based on a monotonically increasing arithmetic progression is novel, simple and efficient, and it does not require complex layer-wise importance calculations or time-consuming searches. The author has proven its effectiveness through theoretical analysis and experimental verification.

2. Significance: The author has conducted extensive experiments to verify the effectiveness of the proposed method, including experiments on LLMs with various architectures and parameters, as well as on multiple tasks. The proposed method has significantly improved the accuracy of sparse LLMs, and the accuracy improvement is obvious compared with the SOTA layer-wise sparsity methods. The proposed method also has a significant improvement effect on multimodal models and vision models, and can also improve the accuracy of compression techniques including quantization. In addition, sparse LLMs have an acceleration effect on CPU and GPU, which significantly improves the usability of sparse LLMs.

3. Clarity: This paper is well-written, with clear logic, a well-organized structure, and rich content.

Weaknesses:

1. Under the setting of 50% sparsity rate, the proposed ATP method has a relatively limited improvement compared to the Uniform baseline, and also has a limited improvement compared to other layer-wise sparsity rate methods.

2. Some implementation details are missing, for example, how is the proposed ATP method applied to CNN, ViT and LLaVA-1.5.

**Questions For Authors:**

1. The author has evaluated LLMs models with a wide range of architectures, but the experimental results of the Mixture of Experts (MoE) model are lacking. However, the MoE model has received increasing attention due to its powerful performance. Can the author evaluate the proposed method on LLMs based on the MoE architecture? I think this would be beneficial for enhancing the contributions of the paper.

2. The author has presented the experimental results of models with more than 7 billion parameters. I'm wondering whether the ATP method can also bring about performance improvements for the smaller LLaMA-3.2-1B/3B models?

3. Sparse LLMs need to rely on specific sparse inference engines (such as DeepSparse and nm-vllm) to achieve acceleration effects. I am curious about the practicality of DeepSparse and nm-vllm. In other words, on which devices can I use these two inference engines to deploy sparse LLMs?

Overall, I think this paper is a good paper and it is worthy of being accepted. However, it lacks some details and experiments. If the author can address the above weaknesses and questions, I will raise my score.

**Relation To Broader Scientific Literature:**

The author claims that the existing layer-wise sparsity methods for LLMs either determine the sparsity rate of each layer by calculating the importance metric of each layer, or obtain the layer-wise sparsity rate through a search method. The importance metric of LLMs is often heuristic and require a great deal of effort to verify their effectiveness, while the search method is very time-consuming. In contrast, the author proposes that the layer-wise sparsity rate of LLMs can be determined through a monotonically increasing arithmetic progression, eliminating the need for cumbersome layer-wise importance calculations or time-consuming search processes. The above scheme is simple and effective. It can quickly determine the sparsity rate of each layer of LLMs and improve the accuracy of sparse LLMs, which is of great significance to the compression community.

**Theoretical Claims:**

Yes. I have checked the proofs of Theorems 3.1-3.5. The author has proven the effect of the sparsity rate on the reconstruction error, the effect of the reconstruction error of the previous layer on that of the next layer, and that the total reconstruction error of the proposed monotonically increasing sparsity rate scheme is smaller than that of any non-monotonically increasing sparsity rate scheme. I think the author's proofs are correct, and I haven't found any issues.

---

> ### Author Rebuttal · Authors · 2025-03-31
>
> **Thanks for your careful review and comments!**
> > Weakness 1 (Smaller improvements for lower sparsity.)
>
> As sparsity decreases, the returns on performance improvements tend to diminish, which is consistent with common patterns in model pruning and compression. However, we believe that our ATP method still demonstrates quite impressive performance:
>
> 1. **Significance of Relative Gains**
>
> While absolute improvements may appear smaller for models with lower sparsity rates, the relative gains remain significant. For 50% sparse LLaMA-7B obtained using Wanda, our ATP method reduced the average zero-shot accuracy loss from 6.03% to 3.37%, showing a 44.11% relative improvement, compared to only 30.85% relative improvement with OWL.
>
> 2. **Inherent Capabilities of Lower Sparsity Models**
>
> At lower sparsity settings, the inherent capabilities of the model are better preserved. For a 50% sparse LLaMA2-7B obtained using Wanda, the average zero-shot accuracy loss is only 3.66%, compared to 26.55% at 70% sparsity, leaving less room for further improvement.
>
> 3. **Alignment with Theoretical Insights**
>
> Wang et al. [1] established scaling laws for sparse LLMs, revealing the relationship between sparsity rate and model performance, indicating that model performance loss is smaller at lower sparsity rates.
>
> Overall, our ATP method maintains effectiveness across all sparsity settings, offering critical stability and performance advantages.
>
> [1] Wang et al. Q-Sparse: All Large Language Models can be Fully Sparsely-Activated.
> > Weakness 2 (Implementation details of CNN, ViT and LLaVA-1.5.)
> 1. Following Wanda, we only sparsify the linear layers within each block of the ConvNeXt, and we use our ATP method to determine the sparsity rate for each block.
>
> 2. For ViT, we use ATP to determine the sparsity rate for each layer. Each layer contains modules such as attention and MLP.  We use Wanda to sparsify linear layers.
>
> 3. For LLaVA-1.5, we only sparsify the Vicuna model within it. Similar to LLaMA, we determine the sparsity rate for each layer and sparsify linear layers.
>
> > Other Comments Or Suggestions (Typos.)
>
> Thanks for this great and detailed comments. We will fix this typo in the final revision.
> > Question 1 (Lack of MoE experiments.)
>
> We use Wanda to obtain 70% sparse MoE model Mixtral-8x7B.
> |Method|Wikitext2 PPL|HellaSwag|Winogrande|BoolQ|OBQA|PIQA|ARC-e|ARC-c|Mean|
> |-|-|-|-|-|-|-|-|-|-|
> |Dense|3.86|64.95|76.09|85.08|35.40|82.49|84.18|57.25|69.35|
> |Uniform|18.22|39.18|60.06|61.75|20.10|70.01|59.38|28.04|48.36|
> |OWL|16.15|40.31|60.54|62.66|21.00|70.40|60.27|29.22|49.20|
> |DSA|16.22|40.11|60.78|61.90|21.10|70.32|60.06|29.30|49.08|
> |AlphaPruning|15.77|40.52|62.12|61.78|21.40|70.89|60.34|29.35|49.49|
> |**ATP**|**14.30**|**41.98**|**63.89**|**62.82**|**22.50**|**71.76**|**61.12**|**30.60**|**50.67**|
>
> Our ATP method performs excellently on sparse MoE models, further demonstrating its generalizability across different model architectures.
> > Question 2 (Lack of Llama-3.2-1B/3B experiments.)
>
> We use SparseGPT to obtain 70% sparse Llama-3.2-1B/3B.
> |Method|Wikitext2 PPL|HellaSwag|Winogrande|BoolQ|OBQA|PIQA|ARC-e|ARC-c|Mean|
> |-|-|-|-|-|-|-|-|-|-|
> |Llama3.2-1B|9.65|47.72|60.46|63.91|26.40|74.48|65.45|31.31|52.82|
> |Uniform|129.24|27.61|51.00|58.56|13.10|57.42|34.10|19.04|37.26|
> |OWL|111.02|28.05|51.20|61.39|13.40|57.78|35.36|19.51|38.10|
> |DSA|121.70|27.93|51.11|59.98|13.10|57.51|35.02|19.31|37.71|
> |AlphaPruning|115.67|28.22|51.38|60.76|13.70|57.64|34.90|19.24|37.98|
> |**ATP**|**94.78**|**28.73**|**52.37**|**62.08**|**14.40**|**58.71**|**35.40**|**20.35**|**38.86**|
> |Llama3.2-3B|7.73|55.28|69.93|73.27|31.00|76.71|74.54|42.24|60.42|
> |Uniform|65.97|30.01|51.78|62.17|14.40|60.66|37.79|19.71|39.50|
> |OWL|53.40|31.73|52.72|62.20|14.60|61.43|40.49|20.39|40.51|
> |DSA|57.81|30.79|53.89|62.65|14.60|60.99|38.90|20.87|40.38|
> |AlphaPruning|60.59|30.15|52.09|62.27|14.80|61.47|38.55|20.39|39.96|
> |**ATP**|**46.64**|**33.66**|**59.32**|**62.49**|**17.10**|**62.29**|**41.53**|**21.19**|**42.51**|
>
> Our ATP method demonstrates consistent improvements across all parameter levels, performing well for models ranging from 1B to 70B parameters.
> > Question 3 (On which devices can sparse LLMs be deployed?)
> 1. Due to the sparsity of weights in unstructured pruning, we must use a specific sparse inference engine to accelerate inference. We use DeepSparse and nm-vllm to accelerate inference on general deployment environments, including CPUs and GPUs.
> 2. For CPUs, DeepSparse supports architectures including: x86 AVX2, AVX-512, AVX-512 VNNI, and ARM v8.2+, which covers most Intel, AMD, and Apple M-series CPUs.
> 3. For GPUs, as long as the device supports CUDA, inference acceleration can be achieved using the nm-vllm. Similarly, if CUDA is supported for other deployment environments, nm-vllm can also be used to achieve acceleration.
>
> **We will incorporate the above responses into the final version. We hope this addresses your concerns. Thank you!**

---

### Official Review · Reviewer_2G3A · 2025-03-12

**Overall Recommendation:** 4

**Summary:**

This work proposes a relatively simple monotonically increasing layer-wise sparsity schedule for LLMs where the layers near the head are more sparse than earlier layers. The method is motivated by a detailed analysis of the effect of increasing sparsity on layerwise reconstruction errors and the propagation of these errors to deeper layers. The authors’ analysis is supported by extensive empirical experiments on one-shot pruned LLMs with a wide variety of leading methods. Across the board, the proposed method yields significant quality benefits compared to the pruning baselines considered and performs on par with bayesian search.

**Claims And Evidence:**

In general the claims made are supported by the empirical data and proofs provided. However, while I am convinced that ATP benefits the one-shot pruning setting, I remain unconvinced that ATP finds the best layer-wise sparsity distribution for LLMs in general after fine-tuning the pruned network. The fine-tuning experiments conducted with LoRA (Section F.8) are not sparsity-preserving and ATP only shows a modest benefit over AlphaPruning in this setting.

**Essential References Not Discussed:**

None noted.

**Experimental Designs Or Analyses:**

* Generally, the experimental designs appeared to be sound.
* However, one instance where I believe the authors could improve their results is to include fine-tuning with a *sparsity-preserving* method. LoRA adapters after fine-tuning yield a dense matrix which will destroy the sparsity introduced by pruning if merged with the original weight matrices – which is the typical approach. As such, I believe an important but missing experiment is to verify whether ATP provides any benefits after sparsity preserving fine-tuning. Examples of such methods include masked fine-tuning (simply masking weights and gradients with the sparsity introduced with pruning) or  sparsity-preserving PEFT methods such as those introduced in [1-3].

[1] W. Huang et al., “Dynamic Low-Rank Sparse Adaptation for Large Language Models,” Feb. 20, 2025, arXiv: arXiv:2502.14816. doi: 10.48550/arXiv.2502.14816.
[2] Y. Hu, J. Zhang, X. Chen, Z. Zhao, C. Li, and H. Chen, “LoRS: Efficient Low-Rank Adaptation for Sparse Large Language Model,” Jan. 15, 2025, arXiv: arXiv:2501.08582. doi: 10.48550/arXiv.2501.08582.
[3] X. Lu, A. Zhou, Y. Xu, R. Zhang, P. Gao, and H. Li, “SPP: Sparsity-Preserved Parameter-Efficient Fine-Tuning for Large Language Models,” May 25, 2024, arXiv: arXiv:2405.16057. doi: 10.48550/arXiv.2405.16057.

**Methods And Evaluation Criteria:**

The method and evaluation criteria are typical for evaluating the performance of sparse LLMs. A wide range of downstream tasks were included as well as a perplexity analysis and performance using Neural Magic inference frameworks.

**Other Comments Or Suggestions:**

None noted.

**Other Strengths And Weaknesses:**

## Strengths:
* Important and timely topic
* Strong empirical results
* Intuitive method with theoretical justification
* The method proposed is efficient and does not require significantly more time or compute than the considered baselines

## Weakness:
* The main weakness is the missing sparsity preserving fine-tuning comparison. Especially since the LoRA results show a convergence in accuracy between ATP and the considered baselines. A crucial finding will be whether ATP is primarily important for improving one-shot accuracy only or if these benefits also extend to the post fine-tuning setting when the sparse mask is fixed prior to fine-tuning.
* While ATP improves the quality of the sparse LLMs compared to baselines, the quality still falls far short of the dense models at moderate sparsities such as 70%. Improving the quality of moderate to highly sparse LLMs remains a challenge, even with ATP.

**Questions For Authors:**

1. How does ATP compare with baselines when fine-tuned with a static mask or a sparsity preserving PEFT method? I would be willing to increase my score if ATP’s benefits remain after static mask fine-tuning.

**Relation To Broader Scientific Literature:**

LLM efficiency is a crucial consideration given their high computational cost. Weight sparsity is one promising avenue to improve LLM performance as most models are memory-bound on current hardware. As such, reducing the number of non-zero elements in the model parameters yields potential performance gains but reducing the amount of HBM I/O and the overall VRAM required (depending on the fine-grained sparse structure and sparsity level).

**Theoretical Claims:**

I reviewed the proofs and did not find any issues.

---

> ### Author Rebuttal · Authors · 2025-04-01
>
> **Thanks for your careful review and comments!**
>
> > Weakness 1 (Missing sparsity preserving fine-tuning results.)
>
> Thank you for your suggestions. The results of fine-tuning 70% sparse LLaMA2-7B obtained by Wanda using LoSA [1], LoRS [2] and SPP [3] are below.
>
> |Method|Fine-tuning|Wikitext2 PPL|HellaSwag|Winogrande|BoolQ|OBQA|PIQA|ARC-e|ARC-c|Mean|
> |-|-|-|-|-|-|-|-|-|-|-|
> |Dense|N.A.|5.12|57.17|68.90|77.74|31.40|78.07|76.39|43.52|61.88|
> |Uniform|LoSA|12.68|43.23|58.55|67.04|23.00|69.07|59.06|28.58|49.79|
> |Uniform|LoRS|13.77|40.11|56.03|65.19|20.70|67.05|55.08|25.04|47.03|
> |Uniform|SPP|12.74|42.51|58.24|66.14|24.50|69.01|59.31|28.05|49.68|
> |OWL|LoSA|11.71|44.02|63.81|69.01|25.10|68.99|58.97|28.67|51.22|
> |OWL|LoRS|13.06|43.01|63.04|69.25|23.00|67.12|54.25|29.54|49.89|
> |OWL|SPP|11.55|44.43|64.25|68.38|24.60|69.02|59.04|29.03|51.25|
> |DSA|LoSA|12.09|43.71|63.01|69.35|24.30|68.07|58.06|28.71|50.74|
> |DSA|LoRS|13.14|43.15|62.89|69.02|22.80|66.80|53.01|29.51|49.59|
> |DSA|SPP|11.99|44.01|63.88|69.01|24.00|68.45|58.37|28.56|50.89|
> |AlphaPruning|LoSA|11.49|44.97|64.09|69.97|25.20|69.01|56.48|28.87|51.22|
> |AlphaPruning|LoRS|12.98|43.47|63.77|69.14|22.40|68.23|54.67|29.09|50.11|
> |AlphaPruning|SPP|11.54|45.05|63.01|68.02|25.40|68.93|57.91|30.55|51.27|
> |ATP|LoSA|**10.68**|45.88|62.95|70.84|25.50|70.98|59.58|29.94|**52.23**|
> |ATP|LoRS|**12.01**|44.31|61.10|68.62|24.90|69.39|59.93|29.42|**51.10**|
> |ATP|SPP|**10.87**|45.19|63.14|68.23|26.40|70.89|61.46|29.44|**52.11**|
>
> We observe that:
> 1. **After Fine-tuning, ATP still Outperforms other Methods**
>
> After fine-tuning using sparsity-preserving methods, the advantages of ATP have been further highlighted. After fine-tuning with LoSA, the perplexity of **ATP is 2.00 lower than Wanda and 0.81 lower than AlphaPruning**. In terms of average zero-shot accuracy, after fine-tuning with LoSA, **ATP achieves 2.44% higher than Wanda and 1.01% higher than AlphaPruning**.
>
> We believe that since ATP better preserves the performance of the one-shot pruned model, the performance of the sparse model is easier to recover through fine-tuning in this case. Therefore, the layer-wise sparsity rate determined by ATP still retains its advantages after fine-tuning with the sparsity-preserving fine-tuning methods and remains the optimal layer-wise sparsity.
>
> 2. **Significance of Relative Gains**
>
> While absolute improvements may appear smaller for fine-tuned models, the relative gains remain significant. For example, compared to dense model, Wanda w. LoSA shows an average zero-shot accuracy loss of 12.09%, while ATP w. LoSA shows a loss of 9.65%, **representing a relative improvement of 20.18%. In contrast, AlphaPruning w. LoSA only achieves a relative improvement of 11.83%.** Considering that the performance gap between sparse and dense LLMs further decreases after fine-tuning, the aforementioned accuracy improvement is likely to translate into more meaningful performance enhancements in practical applications.
>
> **We will include these results in our final version.**
>
> > Weakness 2 (Performance of the sparse model with 70% sparsity has a gap compared to dense model.)
>
> We have observed that performance of 70% sparse LLMs obtained by ATP still underperform compared to dense models. This observation is consistent with the common pattern in model pruning and compression, where high sparsity rates lead to significant accuracy degradation. However, we believe our ATP method still offers the following advantages:
>
> 1. **Pushing the Performance Limits of Sparse LLMs**
>
> Our ATP method addresses the challenge of performance degradation under high sparsity rates, advancing the frontier of sparse LLMs performance and narrowing the gap with dense models, which is beneficial to the community.
>
> 2. **Good Performance at Lower Sparsity Rates**
>
> Under lower sparsity settings, our ATP method further reduces the performance gap between sparse and dense LLMs. For example, the average zero-shot accuracy loss for 50% sparse LLaMA2-13B model obtained using the Wanda method is 3.52%, while our ATP method reduces this loss to 1.07%. This is highly advantageous for the practical deployment of lower-sparsity LLMs.
>
> 3. **Further Narrowing the Gap with Dense Models When Combined with Fine-Tuning**
>
> As demonstrated in Table 16 and response to Weakness 1, we show that combining ATP with PEFT techniques can further improve the accuracy of sparse LLMs. For example, the average zero-shot accuracy loss for 70% sparse LLaMA2-7B model obtained using Wanda is 26.55%. With ATP and LoSA optimizations, this accuracy loss is further reduced to 9.65%. Moreover, sparse LLMs obtained using the ATP method achieve higher accuracy, retaining this advantage even after fine-tuning.
>
> In summary, although there remains a gap between sparse and dense LLMs under high sparsity settings, our ATP method significantly narrows this gap, greatly enhancing the practicality of sparse LLMs.
>
> **Finally, we hope our response has addressed your concerns. Thank you!**

---

> > ### Comment · Reviewer_2G3A · 2025-04-04
> >
> > I thank the authors for the detailed rebuttal; my original concerns have been adequately addressed. At this time I will maintain my original rating.

---

> > > ### Author Response · Authors · 2025-04-08
> > >
> > > Thanks for your acknowledge of our work. Thanks again for your time and effort in reviewing our paper.

---

### Official Review · Reviewer_vXo3 · 2025-03-14

**Overall Recommendation:** 3

**Summary:**

The paper establishes a relationship between sparsity and reconstruction error in pruning LLMs, demonstrating that increased sparsity leads to higher reconstruction error, which propagates through linear layers. The authors support this claim through empirical evaluation on transforms and theoretical analyses on linear layers, showing that pruning strategies that do not account for this effect may result in higher total error, ultimately degrading model performance.

**Claims And Evidence:**

The authors argue that sparsity increases the reconstruction error between the output of a compressed layer and its dense counterpart. This claim is supported by both theoretical analysis of a linear layer and empirical evaluations (Figure 1). Additionally, they suggest that this reconstruction error propagates through the network—again backed by theoretical analysis of a linear layer and empirical evaluation —potentially compounding with each subsequent layer. As a result, the authors propose that per-layer sparsity should account for this propagation, with sparsity increasing monotonically as the layer number increases. To achieve this, they introduce a simple algorithm called APT, which determines layer-wise sparsity using an arithmetic progression. The soundness of the algorithm is verified by undergoing extensive evaluation to confirm its effectiveness. Furthermore, the authors provide theoretical analysis (again focusing on linear layers) to justify the necessity of a monotonically increasing sparsity-per-layer.

I find the empirical evaluations—and the claims based on them—quite strong and comprehensive, even though that idea behind APT is quite simple (i.e. the simplicity actually makes it interesting and compelling). However, my main concern lies with the theoretical analysis. While, to the best of my knowledge, it is correct, it is conducted exclusively on linear layers. Yet, the paper claims from the outset (Section 3.1) to be working with transformers, which incorporate multiple architectural components such as attention mechanisms, nonlinearities, and layer normalization. These elements could significantly influence the theoretical results, making it difficult to assess the practical relevance of these findings for the architectures in question. Moreover, the paper does not address this limitation, nor does it provide any discussion on its implications (See also “Theorems.”). Since theory is posed as one of the main contributions of the paper and it is clear that the authors gave it a lot of thought, I think that this issue of discrepancy (linear layer analysis versus using it to make claims on transformers) should be addressed, since otherwise half of the work seems a little disconnected from the empirical evaluations.

**Essential References Not Discussed:**

[1] Frankle, Jonathan, et al. "Pruning neural networks at initialization: Why are we missing the mark?." arXiv preprint arXiv:2009.08576 (2020).

**Experimental Designs Or Analyses:**

Checked Sections 4.2, 4.3, 4.4, 4.5, 4.6, the experimental design seems sound apart from the fact that I am not sure whether the reported values are averages from repeated experiments and what were the deviations.

**Methods And Evaluation Criteria:**

The majority of the experiments is conducted on the LLama model of various sizes. The studied algorithms for pruning include Wanda and SparseGPT, which are widely-used and well-recognized pruning approaches for zero-shot adaptation. Apart from that, additional experiments on other LLM architectures are provided in the appendix (Table 8). The paper proposes a new scheme for determining sparsity-per-layer densities, and as such compares with other commonly used schemes such as Uniform, OWL, DSA, AlphaPruning apple together with the selected pruning algorithms. In addition Table 1 includes an exploratory algorithm, ALS, and in the Appendix the authors also study some sparsity-per-layer schemes commonly used in non-LLM models (Table 14). In general, I consider the empirical evaluation solid, sound and quite extensive.

**Other Comments Or Suggestions:**

How do we determine that this range is actually "small"? Such a claim shouldn't be based solely on the size of the interval—after all, any interval on RR has the same cardinality as RR. :) I see that the authors conducted a grid search over this parameter in the Appendix, but what I’m really curious about is how sensitive APT is to variations in this hyperparameter. In other words, beyond showing that the optimal β tends to be of the same magnitude across different models, it would be more convincing to demonstrate that small changes in β do not drastically impact performance. This would better support the claim that β is easily tunable.

**Other Strengths And Weaknesses:**

Strengths:
- Work establishes a relationship between the sparsity and the reconstruction error in the pruning of LLMs, studying this aspect both from the empirical and theoretical perspective, showing that increased sparsity  leads to increased reconstruction error, which propagates through layers. Hence schemes that do not address this issue may results in increased total error and hinder the performance of the pruned model.
-  Based on this observation a simple method based on arithmetic linear progression is proposed. The method is easy to use (i.e. quite straightforward use of the established relation), but at the same time produces compelling results in comparison to other sparsity schemes.
-  The work contains numerous experiments on various architectures, as well as modalities and per-layer sparsity schemes that were used even in non-LLM context (Table 14).
- Overall, the paper is well written and easy to follow.

Weaknesses:

Discussed in paragraphs above (epsecially see "Claims and Evidence", "Theorems" and  and “Questions”)

**Questions For Authors:**

1. What is the relevance of theoretical analysis on linear networks when the focus is on transformers? Could you revise Section 3.1 to clarify any potential confusion? Specifically, why introduce transformer layers only to shift to linear layers for theoretical analysis? Additionally, which part of the model is being sparsified? Clarifying these points is crucial to properly validate the theoretical contributions of APT. (See also Clarity and Evidence and Theorems for further discussion.). **Moreover, this is the point that guided my decision on the score of the paper.**

2. Please address the concerns raised in the specific paragraphs above, aside from those already covered in Point 1. For example, the discussion on Relation to Broader Scientific Literature could be expanded to better position the paper within the context of related work. While I find this issue less critical than the concerns in Point 1, addressing it would still enhance the paper’s standing in relation to existing research.

**Relation To Broader Scientific Literature:**

The paper discusses the problem of discovering the optimal sparsity-per-layer budget in pruning methods for LLMs. As such, I believe the work should also discuss related work on the importance of  sparsity-per-layer ratios on the outcome of pruning even for non-LLM models in the related work, eg. [1]

**Theoretical Claims:**

I have checked the correctness of Theorem 3.1, 3.2, Lemma 3.3, and Theorem 3.3, but only skimmed over Theorem 3.5. My general issue with the theorems is that they discuss linear layers (even with no nonlinearity), while the beginning of the Methodology sections promises transformer layers (Section 3.1, first paragraph). This is confusing - see "Claims And Evidence". Furthermore, In Theorem 3.1 and later we assume that the input to the pruned and unpruned network at layer $i$ did not change. In general I believe that any assumptions should be included in the text of the Theorem, not just appear in the proof.

---

> ### Author Rebuttal · Authors · 2025-03-31
>
> **Thanks for your careful review and comments!**
> > Theoretical Claims (Any assumptions should be included in the text of Theorem, not just appear in the proof.)
>
> We will restate Theorem 3.1 as: *When the input is the same, increasing the sparsity of the weights in the $i$-th layer will lead to an increase in the reconstruction error of this layer.*
> > Experimental Designs Or Analyses (Report average and variance of results.)
>
> Following OWL, DSA and AlphaPruning, all experimental results are conducted under a single fixed random seed. We have reported WikiText2 perplexity of 70% sparse LLaMA2-7B obtained by Wanda across five random seeds and different calibration sets. The variance across random seeds is very low, suggesting the robustness of ATP.
> |Method|PPL|
> |-|-|
> |Dense|5.12(±0.00)|
> |Uniform|74.26(±0.10)|
> |OWL|30.38(±0.09)|
> |DSA|63.71(±0.08)|
> |AlphaPruning|28.87(±0.06)|
> |**ATP**|**22.16(±0.05)**|
> > Other Comments Or Suggestions (How do we determine that the $ β$ range is actually "small"?)
>
> 1. In Sec. 3.3, we state that the reasonable range of $β$ is $0<β≤0.019$ for 70% sparse LLaMA3-8B. To find the best $β$, we use grid search with step size of 0.002, needing only 9 searches. We also test smaller step sizes in Table 4 but find no improvement in results. The step size of 0.002 balances search efficiency and performance well. Therefore, we claim that the reasonable range for $β$ is small, where "small" refers to the limited number of searches required.
> 2. We analyze the impact of $β$ on perplexity of sparse model in Figure 4, finding that $β$ significantly affects performance. Figure 2 shows that for lower average sparsity, smaller $β$ values are optimal.
>
> > Question 1 (What is the relevance of theoretical analysis on linear networks when the focus is on transformers?)
>
> In Sec. 3.1, we represent a layer's computation as $\boldsymbol{WX}$, where $\boldsymbol{W}$ is the layer's weight and $\boldsymbol{X}$ is input. A layer includes components such as attention, nonlinearities, and layer normalization. We acknowledge that there are differences between the theoretical analysis based on $\boldsymbol{WX}$ and the actual architecture, given the presence of more complex nonlinear computations in the network. However, we believe our analysis remains reasonable for the following reasons:
> 1. **The theoretical modeling of $\boldsymbol{WX}$ is sufficient to analyze the layer's reconstruction error.**
>
> Our method sparsifies the linear layers in Attention and MLP modules, while other components remain unaffected. These linear layers account for the majority of the parameter count and significantly influence the computation results of the layer. The sparsified linear layers dominate the computation of the reconstruction error for each layer. Although various nonlinear operations exist in the actual architecture, they typically do not fundamentally alter the reconstruction error of each layer and have minimal impact on theoretical analysis. Therefore, we believe modeling the primary computation of a layer as $\boldsymbol{WX}$ is sufficient for analyzing the reconstruction error of that layer. This is also sufficient for us to analyze how reconstruction errors accumulate and propagate across the network.
>
> 2. **Modeling the computation of modules as $\boldsymbol{WX}$ is a common practice in many works.**
>
> [A1] and [A2] simplify the computation of the CONV+BatchNorm+RELU modules in quantized convolutional neural networks as $\boldsymbol{WX}$ when analyzing reconstruction error. This approach of ignoring unnecessary computations and focusing on the core computations is a common practice, which facilitates the derivation of theoretical results.
>
> Thank you for providing valuable insights, which are very important for improving our work. **We will include the above clarification into the final version.**
>
> [A1] Up or down? adaptive rounding for post-training quantization. ICML 2020.
>
> [A2] Solving oscillation problem in post-training quantization through a theoretical perspective. CVPR 2023.
> > Question 2 (Discuss Frankle et al.'s work [1])
>
> Thank you for providing such awesome work!
> 1. Frankle et al. [1] suggest that the effectiveness of initialization pruning mainly depends on each layer's pruning rate, rather than the specific selection of weights within layers. This highlights the importance of layer-wise sparsity rates and supports the value of our work.
> 2. We have discussed and compared various layer-wise sparsity methods, including those for LLMs (methods in Table 1) and CNN/ViT (methods in Table 14). Unlike all previous methods, our ATP method discovers that using a simple monotonically increasing arithmetic progression for layer-wise sparsity can achieve excellent results.
>
> We will include Frankle et al.'s work [1] in our final version.
>
> **We will include the above discussions in our final version. We hope this response has addressed your concerns and kindly ask for a more positive evaluation of our work. Thank you!**

---

> > ### Comment · Reviewer_vXo3 · 2025-04-03
> >
> > Thank you for the response. Your answers mostly cover my concerns. Let me though be a little more clear about my expectations regarding the issue of the relevance of theoretical analysis on linear networks when the focus is on transformers:
> >
> > Your assumption that reconstruction error grows similarly in transformers seems reasonable, especially given the empirical evidence in Figure 1. My main issue is that the paper jumps from the linear case to the transformer case without mentioning those changes or explaining the made approximations.  For instance, Section 3.1 introduces LLM layers (e.g., attention, layer norms) but then abruptly applies a linear approximation in Eq. (1) without stating it as an approximation. The notation also mixes transformer layers with the “channel” convolutional terminology, which is confusing.
> >
> > My point is that I would like the paper to clearly reflect that the theoretical investigations are made on a simplified linear network, and then provide a separate section on how/why does results can be transferred to a transformer network (including  the discussion provided by you in the response above). For instance, I would  suggest restructuring Section 3 as follows:
> >
> > - First, discuss error propagation in linear networks, presenting all relevant theorems and proofs before introducing transformer layers (moving content from Section 3.1's second half, Section 3.2, and Section 3.4).
> > - Next, introduce transformer layers and explicitly state where Equation (1) applies within them (my understanding is that the error is computed on each linear layer in the model, but I did not see such information in Section 3). Provide arguments for transferring insights from linear networks to transformers, backed by empirical results, follow with 3.5.
> >
> > From the theoretical point of view, my (intuitive) concern was that the lower bound in Lemma 3.3 may behave differently due to the attention mechanism, which introduces additional terms like $X_i^T(W_K^TW_Q-\tilde{W_k}^T\tilde{W_Q})X_i$. Would it make sense to analyze these terms (in addition to the restructuring mentioned above) rather than approximating the entire block as a linear projection?
> >
> > On a side note, regarding [A1-A2], approximating a convolutional layer as a linear projection seems more natural than doing so for an attention-based one, given that convolutions can be represented linearly (e.g., via DBT matrices or im2col [B1]).
> >
> > Either way, I slightly increased my score since apart from this “linear analysis vs transformers” point (in which I would be satisfied with a clear clarification made in the text of the paper) the authors have addressed my issues.
> >
> > **References:**
> >
> > [B1] Wang, Jiayun, et al. "Orthogonal convolutional neural networks." Proceedings of the IEEE/CVF conference on computer vision and pattern recognition. 2020.

---

> > > ### Author Response · Authors · 2025-04-08
> > >
> > > **Thanks for your further response.**
> > > > The notation also mixes transformer layers with the “channel” convolutional terminology, which is confusing.
> > >
> > > We apologize for the confusion we have caused. We will revise line 161 from "$c_{in}$ and $c_{out}$ represent the number of input and output **channels**" to "$c_{in}$ and $c_{out}$ represent the number of input and output **feature dimensions**" in our final version.
> > >
> > > > My point is that I would like the paper to clearly reflect that the theoretical investigations are made on a simplified linear network, and then provide a separate section on how/why does results can be transferred to a transformer network (including the discussion provided by you in the response above). For instance, I would suggest restructuring Section 3.
> > >
> > > Thank you for your valuable suggestions. **We will restructure Section 3 in our final version according to your suggestions.** This includes:
> > > 1. Discuss error propagation in linear networks and introduce all relevant theorems and proofs. This part includes the content from the second half of Section 3.1 and Section 3.2.
> > > 2. Add a new Section 3.3 to introduce transformer layers and discuss the relationship between the theoretical analysis of linear networks and the actual transformer network, and elaborate on the rationality of transferring the theoretical analysis based on linear networks to the actual transformer network. This part includes the content from the first half of Section 3.1 and the content we replied during the rebuttal period.
> > > 3. The existing Section 3.3 and 3.4 will be changed to the new Section 3.4 and 3.5 respectively.
> > >
> > > > From the theoretical point of view, my (intuitive) concern was that the lower bound in Lemma 3.3 may behave differently due to the attention mechanism, which introduces additional terms like $X_i^T(W^T_KW_Q-\tilde{W}_K^T\tilde{W}_Q)X_i$. Would it make sense to analyze these terms (in addition to the restructuring mentioned above) rather than approximating the entire block as a linear projection?
> > >
> > > Thank you for your feedback. We understand that your concern was that the lower bound in Lemma 3.3 might behave differently due to the attention mechanism. However, we believe that our formulation and proof of Theorem 3.2 and Lemma 3.3 are reasonable, **as they are supported by empirical evidence**.
> > >
> > > Theorem 3.2 and Lemma 3.3 shows that an increase in the reconstruction error of the previous layer in a sparse LLM usually leads to a further increase in the lower bound of the reconstruction error of the subsequent layer. In practice, this often means that an increase in the reconstruction error of the previous layer will lead to an increase in the reconstruction error of the subsequent layer. We have also observed this phenomenon in the left of Figure 1, where we have plotted the layer-wise reconstruction errors of different layer-wise sparsity methods on the LLaMA2-7B model. We can see that when the reconstruction error of the earlier layers is smaller, the reconstruction error of the subsequent layers is also smaller. Conversely, when the reconstruction error of the earlier layers is larger, the reconstruction error of the subsequent layers is also larger. **The above empirical evidence shows that, despite the presence of the attention mechanism, the lower bound of the reconstruction error still increases, and it is reasonable to approximate the entire Transformer layer as a linear projection in Theorem 3.2 and Lemma 3.3.**
> > >
> > > > On a side note, regarding [A1-A2], approximating a convolutional layer as a linear projection seems more natural than doing so for an attention-based one, given that convolutions can be represented linearly (e.g., via DBT matrices or im2col [B1]).
> > >
> > > Thank you for your feedback. We understand that the linear representations of convolutions (via DBT matrices or im2col) are intuitive. However, we believe that approximating the transformer layer as a linear projection has reasonable justifications.
> > >
> > > For example, [C1] employs Procrustes similarity analysis to discover that the embedding transformations between sequential layers in LLMs such as GPT, LLaMA, OPT, and BLOOM exhibit a near-perfect linear relationship, with linear score is 0.99. This indicates that despite the non-linear operations within Transformer layers, the mapping between adjacent layers can still be approximated as a linear transformation. Therefore, we consider it reasonable and natural to model Transformer layer computations using $\boldsymbol{WX}$.
> > >
> > > We will incorporate the above discussion into the final version. Thank you again for your valuable feedback.
> > >
> > > [C1] Your Transformer is Secretly Linear. ACL 2024 main.
> > >
> > > **Thank you again for your detailed suggestions. They are very helpful for further improving our work, and we will incorporate the above discussion into our final version.**

---

### Decision · Program_Chairs · 2025-05-01

**Decision:**

Accept (spotlight poster)

**Comment:**

The work is clearly motivated, with sparsity in LLMs and specifically the layer-wise sparsity ratio being topics of particular interest. The reviewers commented consistently as to the comprehensiveness of the empirical results and soundness/novelty of the methodology. There were relatively few concerns from the reviewers, perhaps notably concerns on the clarity of the theoretical analysis presented, specifically the relevance of this to real-world Transformer architectures, and some implementation details. The author-reviewer discussion period was very productive (thanks to both the authors and reviewers on this), and I believe it is fair to summarize that the rebuttal appears to have largely addressed the reviewers' concerns. All reviewers recommend an accept post-rebuttal, with one weak accept, and three accepts, and given the discussion I would recommend for acceptance.